# MULTIMEASUREMENT GENERATIVE MODELS

**Saeed Saremi[1, 2] & Rupesh Kumar Srivastava[1]**
[1]NNAISENSE Inc.
[2]Redwood Center, UC Berkeley
{saeed, rupesh}@nnaisense.com

## ABSTRACT

We formally map the problem of sampling from an unknown distribution with a density in $\mathbb{R}^d$ to the problem of learning and sampling a smoother density in $\mathbb{R}^{Md}$ obtained by convolution with a fixed factorial kernel: the new density is referred to as *M-density* and the kernel as *multimeasurement noise model* (MNM). The M-density in $\mathbb{R}^{Md}$ is smoother than the original density in $\mathbb{R}^d$, easier to learn and sample from, yet for large $M$ the two problems are mathematically equivalent since clean data can be estimated exactly given a multimeasurement noisy observation using the Bayes estimator. To formulate the problem, we derive the Bayes estimator for Poisson and Gaussian MNMs in closed form in terms of the unnormalized M-density. This leads to a simple least-squares objective for learning parametric energy and score functions. We present various parametrization schemes of interest including one in which studying Gaussian M-densities directly leads to *multidenoising autoencoders*—this is the first theoretical connection made between denoising autoencoders and empirical Bayes in the literature. Samples in $\mathbb{R}^d$ are obtained by walk-jump sampling (Saremi & Hyvärinen, 2019) via underdamped Langevin MCMC (walk) to sample from M-density and the multimeasurement Bayes estimation (jump). We study permutation invariant Gaussian M-densities on MNIST, CIFAR-10, and FFHQ-256 datasets, and demonstrate the effectiveness of this framework for realizing fast-mixing stable Markov chains in high dimensions.

## 1 INTRODUCTION

Consider a collection of i.i.d. samples $\{x_i\}_{i=1}^n$, assumed to have been drawn from an *unknown* distribution with density $p_X$ in $\mathbb{R}^d$. An important problem in probabilistic modeling is the task of drawing independent samples from $p_X$, which has numerous potential applications. This problem is typically approached in two phases: approximating $p_X$, and drawing samples from the approximated density. In unnormalized models the first phase is approached by learning the energy function $f_X$ associated with the Gibbs distribution $p_X \propto \exp(-f_X)$, and for the second phase one must resort to Markov chain Monte Carlo methods, such as Langevin MCMC, which are typically very slow to mix in high dimensions. MCMC sampling is considered an "art" and we do not have black box samplers that converge fast and are stable for complex (natural) distributions. The source of the problem is mainly attributed to the fact that the energy functions of interest are typically highly nonconvex.

A broad sketch of our solution to this problem is to model a *smoother* density in an M-fold expanded space. The new density denoted by $p(\mathbf{y})$, called M-density, is defined in $\mathbb{R}^{Md}$, where the boldfaced $\mathbf{y}$ is a shorthand for $(y_1, \ldots, y_M)$. M-density is smoother in the sense that its marginals $p_m(y_m)$ are obtained by the convolution $p_m(y_m) = \int p_m(y_m|x)p(x)dx$ with a smoothing kernel $p_m(y_m|x)$ which for most of the paper we take to be the isotropic Gaussian:

$$Y_m = X + N(0, \sigma_m^2 I_d).$$

Although we bypass learning $p(x)$, the new formalism allows for generating samples from $p(x)$ since $X$ can be estimated exactly given $\mathbf{Y} = \mathbf{y}$ (for large $M$). To give a physical picture, our approach here is based on "taking apart" the complex manifold where the random variable $X$ is concentrated in $\mathbb{R}^d$ and mapping it to a smoother manifold in $\mathbb{R}^{Md}$ where $\mathbf{Y} = (Y_1, \ldots, Y_M)$ is now concentrated.

Smoothing a density with a kernel is a technique in nonparametric density estimation that goes back to Parzen (1962). In kernel density estimation, the estimator of $p(x)$ is obtained by convolving the

empirical measure with a kernel. In that methodology, the kernel bandwidth ($\sigma$, for Gaussian kernels) is adjusted to estimate $p(x)$ in $\mathbb{R}^d$ given a collection of independent samples $\{x_i\}_{i=1}^n$. This estimator, like most nonparametric estimators, suffers from a severe curse of dimensionality (Wainwright, 2019). But what if the kernel bandwidth is *fixed*: how much easier is the problem of estimating $p(y)$? This question is answered in (Goldfeld et al., 2020), where they obtained the rate of convergence $e^{O(d)}n^{-1/2}$ (measured using various distances) in remarkable contrast to the well-known $n^{-1/d}$ rate for estimating $p(x)$. This nonparametric estimation result is not directly relevant here, but it formalizes the intuition that learning $p(y) = \int p(y|x)p(x)dx$ is a lot simpler than learning $p(x)$.

With this motivation, we start with an introduction to the problem of learning unnormalized $p(y)$, based on independent samples from $p(x)$. This problem was formulated by Vincent (2011) using score matching (Hyvärinen, 2005). It was approached recently with the more fundamental methodology of empirical Bayes (Saremi & Hyvärinen, 2019). The idea is to use the Bayes estimator of $X$ given $Y = y$, the study of which is at the root of the *empirical Bayes approach to statistics* (Robbins, 1956), in a least-squares objective. This machinery builds on the fact that the estimator $\widehat{x}(y) = \mathbb{E}[X|Y = y]$ can be expressed in closed form in terms of unnormalized $p(y)$ (Sec. 3.1). For Gaussian kernels, the learning objective arrived at in (Saremi & Hyvärinen, 2019) is identical (up to a multiplicative constant) to the denoising score matching formulation (Vincent, 2011), but with new insights rooted in empirical Bayes which is *the* statistical framework for denoising.

*The main problem with the empirical Bayes methodology is that $p(x|y)$ remains unknown and cannot be sampled from.* The estimator $\widehat{x}(y) = \mathbb{E}[X|Y = y]$ can be computed, but the concentration of the posterior $p(x|y)$ around the mean is not in our control. Our solution to this problem starts with an observation that is very intuitive from a Bayesian perspective: one can sharpen the posterior by simply taking more independent noisy measurements. This scheme is formalized by replacing $p(y|x)$ with the factorial kernel $p(\mathbf{y}|x)$:

$$p(\mathbf{y}|x) = \prod_{m=1}^{M} p_m(y_m|x), \text{ where } \mathbf{y} = (y_1, \ldots, y_M), \tag{1}$$

which we name **multimeasurement noise model** (MNM). Now, the object of interest is a different density which we call **M-density** obtained by convolving $p(x)$ with the factorial kernel:

$$p(\mathbf{y}) = \int p(\mathbf{y}|x)p(x)dx. \tag{2}$$

This formally maps the original problem of drawing samples from $p(x)$ to drawing samples from $p(\mathbf{y})$ for *any* fixed noise level since the estimator of $X$ given $\mathbf{Y} = \mathbf{y}$ is asymptotically exact. We quantify this for Gaussian MNMs using the plug-in estimator (the empirical mean of measurements).

*Smooth & Symmetric!* Consider Gaussian MNMs with equal noise level $\sigma$ in the regime of large $\sigma$, large $M$ such that $\sigma\sqrt{d/M}$ is "small".[1] In that regime, the complex manifold associated with the data distribution is mapped to a very smooth symmetric manifold in a much higher dimensional space. The original manifold can be reconstructed via a *single step* by computing $\widehat{x}(\mathbf{y})$. Due to equal noise levels, the manifold associated with M-density is symmetric under the permutation group:

$$p(y_1, \ldots, y_M) = p(y_{\pi(1)}, \ldots, y_{\pi(M)}), \tag{3}$$

where $\pi$ is a permutation of indices (Fig. 1). Although we develop a general methodology for studying M-densities, in the later part of the paper we focus on permutation invariant Gaussian M-densities.

The paper is organized as follows. In Sec. 2, we derive Bayes estimators for Poisson and Gaussian MNMs. In Sec. 3, we present the least-squares objective for learning Gaussian M-densities. We also give a weaker formulation of the learning objective based on score matching. Sec. 4 is devoted to the important topic of parametrization, where we introduce **multidenoising autoencoder** (MDAE) in which we formally connect M-densities to the DAE literature. DAEs have never been studied for factorial kernels and the emergence of MDAE as a generative model should be of wide interest. In addition, we introduce **metaencoder** formulated in an unnormalized latent variable model, which is mainly left as a side contribution. In Sec. 5, we present the sampling algorithm used in the paper. In Sec. 6, we present our experiments on MNIST, CIFAR-10, and FFHQ-256 datasets which were focused on permutation invariant M-densities. The experiments are mainly of qualitative nature demonstrating the effectiveness of this method in generating fast mixing and very long Markov chains in high dimensions. Related works are discussed in Sec. 7, and we finish with concluding remarks.

---

[1]The regime $\sigma\sqrt{d/M} \ll 1$ is obtained in our analysis of the highly suboptimal plug-in estimator (Sec. 2.3).

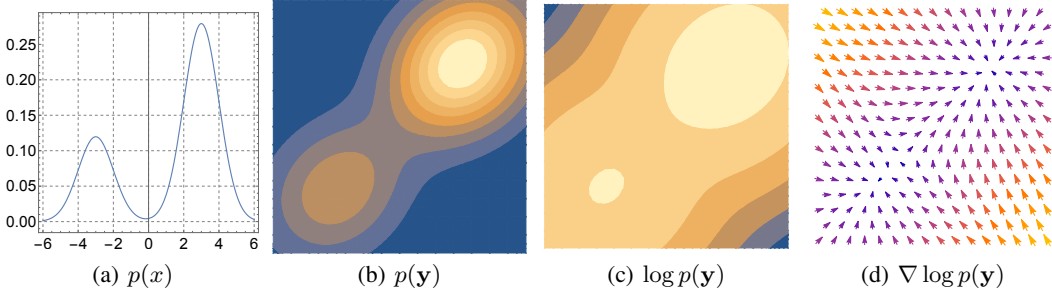

|  |  |  |  |
| :---: | :---: | :---: | :---: |
| (a) $p(x)$ | (b) $p(\mathbf{y})$ | (c) $\log p(\mathbf{y})$ | (d) $\nabla \log p(\mathbf{y})$ |

Figure 1: (*M-density*) (a) A mixture of Gaussian in 1d. (b,c,d) The M-density ($M = 2$, $\sigma_1 = \sigma_2$), the corresponding log-density and score function are visualized (based on calculations in Appendix A).

**Notation.** The subscripts are dropped from densities and energy functions when it is clear from their arguments: $p(\mathbf{y}) = p_{\mathbf{Y}}(\mathbf{y}), p(\mathbf{y}|x) = p_{\mathbf{Y}|X=x}(\mathbf{y}), f(\mathbf{y}) = f_{\mathbf{Y}}(\mathbf{y})$, etc. Bold fonts are reserved for multimeasurement random variables: $\mathbf{Y} = (Y_1, \ldots, Y_M)$. The following are shorthand notations: $[M] = \{1, \ldots, M\}$ and $\nabla_m = \nabla_{y_m}$. Throughout, $\nabla$ is the gradient with respect to inputs (in $\mathbb{R}^{Md}$), not parameters. The following convention is used regarding parametric functions: $f_\theta(\cdot) = f(\cdot; \theta)$. Different parametrization schemes come with a different set of parameters, the collection of which we denote by $\theta$. For all the datasets used in the paper, $X$ takes values in the hypercube $[0, 1]^d$.

## 2 FORMALISM: MULTIMEASUREMENT BAYES ESTIMATORS

This work is based on generalizing the empirical Bayes methodology to MNMs. It is well known that the least-squares estimator of $X$ given $\mathbf{Y} = \mathbf{y}$ (for any noise model) is the Bayes estimator:

$$\widehat{x}(\mathbf{y}) = \frac{\int x p(\mathbf{y}|x) p(x) dx}{\int p(\mathbf{y}|x) p(x) dx}. \tag{4}$$

Next we study this estimator à la Robbins (1956) for Poisson (the Poisson kernel was the first example studied in 1956) and Gaussian MNMs. In both cases the estimator $\widehat{x}(\mathbf{y})$ is derived to be a functional of the joint density $p(\mathbf{y})$. In addition, $\widehat{x}(\mathbf{y})$ is invariant to scaling $p(\mathbf{y})$ by a constant, therefore *one can ignore the partition function in this estimation problem*. This is the main appeal of this formalism. Analytical results for Poisson MNMs are included to demonstrate the generality of the new formalism, but we will not pursue it as a generative model in our experiments for technical reasons due to the challenges regarding sampling discrete distributions in high dimensions (see Remark 7).

### 2.1 POISSON MNM

Let $X$ be a random variable taking values in $\mathbb{R}_+$. The Poisson MNM is defined by:

$$p(\mathbf{y}|x) = e^{-Mx} \prod_{l=1}^{M} \frac{x^{y_l}}{y_l!}, \ y_l \in \mathbb{N}.$$

The numerator in r.h.s. of Eq. 4 is computed next. The measurement index $m$ below is an arbitrary index in $[M]$ used for absorbing $x$ such that $xp(\mathbf{y}|x)$ has the same functional form as $p(\mathbf{y}|x)$:

$$\int x p(\mathbf{y}|x) p(x) dx = \int e^{-Mx} (y_m + 1) \frac{x^{(y_m+1)}}{(y_m + 1)!} \prod_{l \neq m} \frac{x^{y_l}}{y_l!} p(x) dx = (y_m + 1) p(\mathbf{y} + \mathbf{1}_m),$$

where $\mathbf{1}_m$ is defined as a vector whose component $l$ is $\delta_{ml}$. Using Eq. 4, it immediately follows

$$\widehat{x}(\mathbf{y}) = (y_m + 1) \frac{p(\mathbf{y} + \mathbf{1}_m)}{p(\mathbf{y})}, \ m \in [M]. \tag{5}$$

We emphasize that the dependency of $\widehat{x}(\mathbf{y})$ on the *noise channel* (measurement index) $m$ that appears on the right hand side of the expression above is only an artifact of the calculation (we observe this again for Gaussian MNMs). The result above holds for any measurement index $m \in [M]$, therefore

$$(y_m + 1) p(\mathbf{y} + \mathbf{1}_m) = (y_{m'} + 1) p(\mathbf{y} + \mathbf{1}_{m'}) \text{ for all } m, m' \in [M].$$

**Example.** We can derive the estimator $\widehat{x}(\mathbf{y})$ analytically for $p(x) = e^{-x}$. We first derive $p(\mathbf{y})$:[2]

$$p(\mathbf{y}) = \frac{(\sum_l y_l)!}{\prod_l y_l!}(M+1)^{-1-\sum_l y_l},$$

where the sums/products are over the measurement indices $l \in [M]$. Using Eq. 5, it follows

$$\widehat{x}(\mathbf{y}) = (y_m + 1)\frac{p(\mathbf{y} + \mathbf{1}_m)}{p(\mathbf{y})} = (y_m + 1)\frac{\sum_l y_l + 1}{y_m + 1}\frac{1}{M+1} = \frac{\sum_l y_l + 1}{M+1}$$

As expected one arrives at the same result by computing Eq. 5 for any measurement index $m$.

## 2.2 GAUSSIAN MNM

Let $X$ be a random variable in $\mathbb{R}^d$. The Gaussian MNM is defined by:

$$p(\mathbf{y}|x) = \prod_{m=1}^{M} p_m(y_m|x), \text{ where } p_m(y_m|x) = \mathcal{N}(y_m; x, \sigma_m^2 I_d). \tag{6}$$

It follows (as in the Poisson MNM, $m$ in the equation below is an arbitrary index in $[M]$):

$$\sigma_m^2 \nabla_m p(\mathbf{y}|x) = (x - y_m)p(\mathbf{y}|x).$$

We multiply both sides of the above expression by $p(x)$ and integrate over $x$. The derivative $\nabla_m$ (with respect to $y_m$) and the integration over $x$ commute, and using Eq. 4 it follows

$$\sigma_m^2 \nabla_m p(\mathbf{y}) = \widehat{x}(\mathbf{y})p(\mathbf{y}) - y_m p(\mathbf{y}),$$

which we simplify by dividing both sides by $p(\mathbf{y})$:

$$\widehat{x}(\mathbf{y}) = y_m + \sigma_m^2 \nabla_m \log p(\mathbf{y}), \ m \in [M]. \tag{7}$$

This expression is the generalization of the known result due to Miyasawa (1961). As in the Poisson MNM, the result above holds for any $m \in [M]$, therefore:

$$y_m + \sigma_m^2 \nabla_m \log p(\mathbf{y}) = y_{m'} + \sigma_{m'}^2 \nabla_{m'} \log p(\mathbf{y}) \text{ for all } m, m' \in [M]. \tag{8}$$

**Example.** We also studied the M-density for Gaussian MNMs analytically. The calculations are insightful and give more intuitions on the new formalism. We refer to **Appendix A** for the results.

**Remark 1** ($\widehat{x}_\theta^{(m)}(\mathbf{y})$ notation). *For parametric M-densities, Eq. 8 is in general an approximation. We use the superscript $m$ to emphasize that there are $M$ ways to compute the parametric estimator:*

$$\widehat{x}_\theta^{(m)}(\mathbf{y}) = y_m + \sigma_m^2 \nabla_m \log p_\theta(\mathbf{y})$$

**Remark 2** ($\sigma \otimes M$ notation). *Gaussian MNMs with the constant noise level $\sigma$ are denoted by $\sigma \otimes M$. For $\sigma \otimes M$ models, the M-density $p(\mathbf{y})$ (resp. the score function $\nabla \log p(\mathbf{y})$) is invariant (resp. equivariant) with respect to the permutation group $\pi : [M] \to [M]$. See Fig. 1 for an illustration.*

## 2.3 CONCENTRATION OF THE PLUG-IN ESTIMATOR

The plug-in estimator of $X$ is the empirical mean of the multiple noisy measurements we denote by $\widehat{x}_{\text{mean}}(\mathbf{y}) = M^{-1} \sum_m y_m$. This estimator is highly suboptimal but its analysis in high dimensions for Gaussian MNMs is useful. Due to the concentration of measure phenomenon (Tao, 2012) we have

$$\left\| x - \widehat{x}_{\text{mean}}(\mathbf{y}) \right\|_2 \approx \sigma_{\text{eff}}\sqrt{d}, \tag{9}$$

where $\sigma_{\text{eff}}$ ("eff" is for effective) is given by

$$\sigma_{\text{eff}} = \frac{1}{M}\left(\sum_{m=1}^{M} \sigma_m^2\right)^{1/2}. \tag{10}$$

The calculation is straightforward since $y_m = x + \varepsilon_m$, where $\varepsilon_m \sim N(0, \sigma_m^2 I_d)$. It follows: $x - \widehat{x}_{\text{mean}}(\mathbf{y}) = -M^{-1}\sum_m \varepsilon_m$ which has the same law as $N(0, \sigma_{\text{eff}}^2 I_d)$. This calculation shows that the estimator of $X$ in $\mathbb{R}^d$ concentrates at the true value at a worst-case rate $O(\sqrt{d/M})$ (consider replacing the sum in Eq. 10 by $M\sigma_{\text{max}}^2$). This analysis is very conservative, as it ignores the correlations between the components of $\mathbf{y}$ in $\mathbb{R}^{Md}$ (within and across noise channels), and one expects a (much) tighter concentration for the *optimal* estimator. In the next section, we present an algorithm to learn the multimeasurement Bayes estimator based on independent samples from $p(x)$.

---

[2]The derivation is straightforward using $\int_0^\infty e^{-\alpha x} x^\beta dx = \alpha^{-1-\beta}\beta!$ for $\alpha > 0, \beta \in \mathbb{N}$.

## 3 LEARNING GAUSSIAN M-DENSITIES

### 3.1 NEURAL EMPIRICAL BAYES

In this section, we focus on learning Gaussian M-densities using the empirical Bayes formalism. We closely follow the approach taken by Saremi & Hyvärinen (2019) and extend it to M-densities. The power of empirical Bayes lies in the fact that it is formulated in the absence of any clean data. This is reflected by the fact that $\widehat{x}(y)$ is expressed in terms of $p(y)$ which can in principle be estimated without observing samples from $p(x)$ (Robbins, 1956); we generalized that to factorial kernels in Sec. 2. What if we *start* with independent samples $\{x_i\}_{i=1}^n$ from $p(x)$ and our goal is to *learn* $p(\mathbf{y})$?

A key insight in *neural empirical Bayes* was that the empirical Bayes machinery can be turned on its head in the form of a Gedankenexperiment (Saremi & Hyvärinen, 2019): we can draw samples (indexed by $j$) from the factorial kernel $\mathbf{y}_{ij} \sim p(\mathbf{y}|x_i)$ and feed the noisy data to the empirical Bayes "experimenter" (the word used in 1956). The experimenter's task (our task!) is to estimate $X$, but since we observe $X = x_i$, the squared $\ell_2$ norm $\left\|x_i - \widehat{x}_\theta(\mathbf{y}_{ij})\right\|_2^2$ serves as a signal to learn $p(\mathbf{y})$, and also $\widehat{x}(\mathbf{y})$ for unseen noisy data (as a reminder the Bayes estimator is the least-squares estimator).

Next, we present the least-squares objective to learn $p(\mathbf{y}) \propto e^{-f(\mathbf{y})}$ for Gaussian M-densities. It is important to note that $\widehat{x}(\mathbf{y})$ is expressed in terms of *unnormalized* $p(\mathbf{y})$, without which we must estimate the partition function (or its gradient) during learning. Here, we can choose any expressive family of functions to parametrize $\widehat{x}(\mathbf{y})$ which is key to the success of this framework. Using our formalism (Sec. 2.2), the Bayes estimator takes the following parametric form (see Remark 1):

$$\widehat{x}_\theta^{(m)}(\mathbf{y}) = y_m - \sigma_m^2 \nabla_m f_\theta(\mathbf{y}), \ m \in [M]. \tag{11}$$

There are therefore $M$ least-squares learning objectives

$$\mathcal{L}^{(m)}(\theta) = \mathbb{E}_{(x,\mathbf{y})} \mathcal{L}^{(m)}(x, \mathbf{y}; \theta), \ \text{where} \ \mathcal{L}^{(m)}(x, \mathbf{y}; \theta) = \left\|x - \widehat{x}_\theta^{(m)}(\mathbf{y})\right\|_2^2 \tag{12}$$

that in principle need to be minimized simultaneously, since as a corollary of Eq. 8 we have:

$$\mathcal{L}^{(m)}(x, \mathbf{y}; \theta^*) \approx \mathcal{L}^{(m')}(x, \mathbf{y}; \theta^*) \ \text{for all} \ m, m' \in [M], \ x \in \mathbb{R}^d, \ \mathbf{y} \in \mathbb{R}^{Md}. \tag{13}$$

The balance between the $M$ losses can be enforced during learning by using a softmax-weighted sum of them in the learning objective, effectively weighing the higher losses more in each update. However, in our parametrization schemes, coming up, simply taking the mean of the $M$ losses as the learning objective proved to be sufficient for a balanced learning across all the noise channels:

$$\mathcal{L}(\theta) = \frac{1}{M} \sum_{m=1}^{M} \mathcal{L}^{(m)}(\theta). \tag{14}$$

The above learning objective is the one we use in the remainder of the paper.

### 3.2 DENOISING SCORE MATCHING

One can also study M-densities using score matching with the following objective (Hyvärinen, 2005):

$$\mathcal{J}(\theta) = \mathbb{E}_{\mathbf{y}} \left\| - \nabla_{\mathbf{y}} f_\theta(\mathbf{y}) - \nabla_{\mathbf{y}} \log p(\mathbf{y}) \right\|_2^2. \tag{15}$$

In **Appendix B**, we show that the score matching learning objective is equal (up to an additive constant independent of $\theta$) to the following *multimeasurement denoising score matching* (MDSM) objective:

$$\mathcal{J}(\theta) = \sum_{m=1}^{M} \mathcal{J}_m(\theta), \ \text{where} \ \mathcal{J}_m(\theta) = \mathbb{E}_{(x,\mathbf{y})} \left\| - \nabla_m f_\theta(\mathbf{y}) + \frac{y_m - x}{\sigma_m^2} \right\|_2^2. \tag{16}$$

This is a simple extension of the result by Vincent (2011) to Gaussian MNMs. The MDSM objective and the empirical Bayes' (Eq. 14) are identical (up to a multiplicative constant) for $\sigma \otimes M$ models.

**Remark 3.** *Compared to neural empirical Bayes (NEB), denoising score matching (DSM) takes a very different approach regarding learning M-densities. DSM starts with score matching (Eq. 15). NEB starts with deriving the Bayes estimator of $X$ given $\mathbf{Y} = \mathbf{y}$ for a known kernel $p(\mathbf{y}|x)$ (Eq. 7). NEB is a stronger formulation in the sense that two goals are achieved at once: learning M-density and learning $\widehat{x}(\mathbf{y})$. What remains unknown in DSM is the latter, and knowing the estimator is key here. Without it, we cannot draw a formal equivalence between $p_X$ and $p_{\mathbf{Y}}$ (see Sec. 1). We return to this discussion at the end of the paper from a different angle with regards to denoising autoencoders.*

## 4 PARAMETRIZATION SCHEMES

We present three parametrization schemes for modeling Gaussian M-densities. Due to our interest in $\sigma \otimes M$ models we switch to a lighter notation. Here, the learning objective (Eq. 14) takes the form

$$\mathcal{L}(\theta) = \frac{1}{M} \mathbb{E}_{(x,\mathbf{y}) \sim p(\mathbf{y}|x)p(x)} \big\| x \otimes M - \mathbf{y} + \sigma^2 \nabla f_\theta(\mathbf{y}) \big\|_2^2, \tag{17}$$

where $x \otimes M$ denotes $(x, \ldots, x)$ repeated $M$ times. Parametrization schemes fall under two general groups: *multimeasurement energy model* (MEM) and *multimeasurement score model* (MSM). In what follows, MDAE is an instance of MSM, MEM$^2$ & MUVB are (closely related) instances of MEM.

### 4.1 MDAE (MULTIDENOISING AUTOENCODER)

The rigorous approach to learning Gaussian M-densities is to parametrize the energy function $f$. In that parametrization, $\nabla f_\theta$ is computed with automatic differentiation and used in the objective (Eq. 17). That is a computational burden, but it comes with a major advantage as the learned score function is guaranteed to be a *gradient field*. The direct parametrization of $\nabla f$ is problematic due to this requirement analyzed by (Saremi, 2019); see also (Salimans & Ho, 2021) for a recent discussion. Putting that debate aside, in MDAE we parametrize the score function explicitly $\mathbf{g}_\theta : \mathbb{R}^{Md} \to \mathbb{R}^{Md}$. Then, $\mathbf{g}_\theta$ replaces $-\nabla f_\theta$ in Eq. 17. In particular, we consider the following reparametrization:

$$\mathbf{g}_\theta(\mathbf{y}) := (\boldsymbol{\nu}_\theta(\mathbf{y}) - \mathbf{y})/\sigma^2, \tag{18}$$

simply motivated by the fact we would like to cancel $\mathbf{y}$ in Eq. 17, otherwise the loss starts at very high values at the initialization. This is especially so in the regime of large $\sigma$, $M$, and $d$. It follows:

$$\mathcal{L}(\theta) = M^{-1} \mathbb{E}_{(x,\mathbf{y}) \sim p(\mathbf{y}|x)p(x)} \big\| x \otimes M - \boldsymbol{\nu}_\theta(\mathbf{y}) \big\|_2^2. \tag{19}$$

This is very intriguing and it is worth taking a moment to examine the result: modeling M-densities is now formally mapped to denoising multimeasurement noisy data *explicitly*. It is a DAE loss with a multimeasurement twist. Crucially, due to our empirical Bayes formulation, the MDAE output $\boldsymbol{\nu}_\theta(\mathbf{y})$ becomes a parametrization of the Bayes estimator(s) (combine Eq. 18 and Eq. 11):

$$\widehat{x}^{(m)}(\mathbf{y}; \theta) = \nu_m(\mathbf{y}; \theta). \tag{20}$$

This makes a strong theoretical connection between our generalization of empirical Bayes to factorial kernels and a new learning paradigm under *multidenoising autoencoders*, valid for any noise level $\sigma$.

### 4.2 MEM$^2$ (MEM with a QUADRATIC form with an optional METAENCODER)

Can we write down an energy function associated with the score function in Eq. 18? The answer is no, since $\mathbf{g}_\theta$ is not a gradient field in general, but we can try the following ($\theta = (\eta, \zeta)$):

$$f_\theta(\mathbf{y}) := \frac{1}{2\sigma^2} \big\| \mathbf{y} - \boldsymbol{\nu}_\eta(\mathbf{y}) \big\|_2^2 + h_\zeta(\mathbf{y}, \boldsymbol{\nu}_\eta(\mathbf{y})). \tag{21}$$

Ignoring $h_\zeta$ for now, by calculating $-\nabla f_\theta$ we do get both terms in Eq. 18, plus other terms. The function $h_\zeta$ which we call *metaencoder* is optional here. Intuitively, it captures the "higher order interactions" between $\mathbf{y}$ and $\boldsymbol{\nu}$, beyond the quadratic term (see below for another motivation). The metaencoder is implicitly parametrized by $\eta$ (via $\boldsymbol{\nu}_\eta$), while having its own set of parameters $\zeta$.

### 4.3 MUVB (MULTIMEASUREMENT UNNORMALIZED VARIATIONAL BAYES)

The expression above (Eq. 21) is a simplification of an unnormalized latent variable model that one can set up, where we take the variational free energy to be the energy function. This builds on recent studies towards bringing together empirical Bayes and variational Bayes (Saremi, 2020a;b). We outline the big picture here and refer to **Appendix C** for details. Latent variable models operate on the principle of maximum likelihood, where one is obliged to have normalized models. Since our model is unnormalized we consider setting up a latent variable model with unnormalized conditional density. Essentially both terms in Eq. 21 arise by considering ($z$ is the vector of latent variables)

$$p_{(\eta,\zeta)}(\mathbf{y}|z) \propto \exp\left( -\frac{1}{2\sigma^2} \big\| \mathbf{y} - \boldsymbol{\nu}_\eta(z) \big\|_2^2 - h_\zeta(\mathbf{y}, \boldsymbol{\nu}_\eta(z)) \right), \tag{22}$$

named *metalikelihood* which further underscores the fact that it is unnormalized. The full expression for the energy function also involves the posterior $q_\phi(z|\mathbf{y})$. As a remark, note that what is shared between all three parametrization schemes is $\boldsymbol{\nu}$, although vastly different in how $\widehat{x}(\mathbf{y})$ is computed.

## 5 SAMPLING ALGORITHM

Our sampling algorithm is an adaptation of *walk-jump sampling* (WJS) (Saremi & Hyvärinen, 2019). We run an MCMC algorithm to sample M-density by generating a Markov chain of multimeasurement noisy samples. This is the *walk* phase of WJS schematized by the dashed arrows in Fig. 2. At *arbitrary* discrete time $k$, clean samples are generated by simply computing $\widehat{x}(\mathbf{y}_k)$ ($\theta$ is dropped for a clean notation). This is the *jump* phase of WJS schematized by the solid arrow in Fig. 2. What is appealing about *multimeasurement generative models* is the fact that for large $M$, $p(x|\mathbf{y}_k)$ is highly concentrated around its mean $\widehat{x}(\mathbf{y}_k)$, therefore this scheme is an exact sampling scheme—this was in fact our original motivation to study M-densities. For sampling the M-density (the walk phase), we consider Langevin MCMC algorithms that are based on discretizing the underdamped Langevin diffusion:

$$d\mathbf{v}_t = -\gamma\mathbf{v}_t dt - u\nabla f(\mathbf{y}_t)dt + (\sqrt{2\gamma u})d\mathbf{B}_t,$$
$$d\mathbf{y}_t = \mathbf{v}_t dt. \tag{23}$$

Here $\gamma$ is the friction, $u$ the inverse mass, and $\mathbf{B}_t$ the standard Brownian motion in $\mathbb{R}^{Md}$. Discretizing the Langevin diffusion and their analysis are challenging problems due to the non-smooth nature of the Brownian motion (Mörters & Peres, 2010). There has been a significant progress being made however in devising and analyzing Langevin MCMC algorithms, e.g. Cheng et al. (2018) introduced an algorithm with a mixing time that scales as $O(\sqrt{d})$ in a notable contrast to the best known $O(d)$ for MCMC algorithms based on overdamped Langevin diffusion. *The dimension dependence of the mixing time is of great interest here since we expand the dimension M-fold.* To give an idea regarding the dimension, for the $4 \otimes 8$ model on FFHQ-256, $Md \approx 10^6$. We implemented (Cheng et al., 2018, Algorithm 1) in this paper which to our knowledge is its first use for generative modeling. In addition, we used a Langevin MCMC algorithm due to Sachs et al. (2017), also used by Arbel et al. (2020). Note, in addition to $\gamma$ and $u$, Langevin MCMC requires $\delta$, the step size used for discretizing Eq. 23.

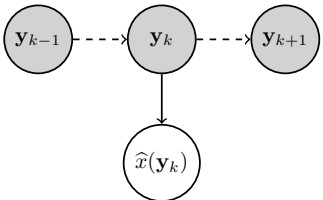

Figure 2: (*WJS schematic*) In this schematic, the Langevin *walk* is denoted by the dashed arrow. The *jump* is denoted by the solid arrow which is deterministic. The jumps in WJS are asynchronous (Remark 5). In presenting long chains in the paper we show jumps at various frequencies denoted by $\Delta k$ (Remark 6). We use the same MCMC parameters for all noise channels due to permutation symmetry in $\sigma \otimes M$ models. See **Appendix D** for more details.

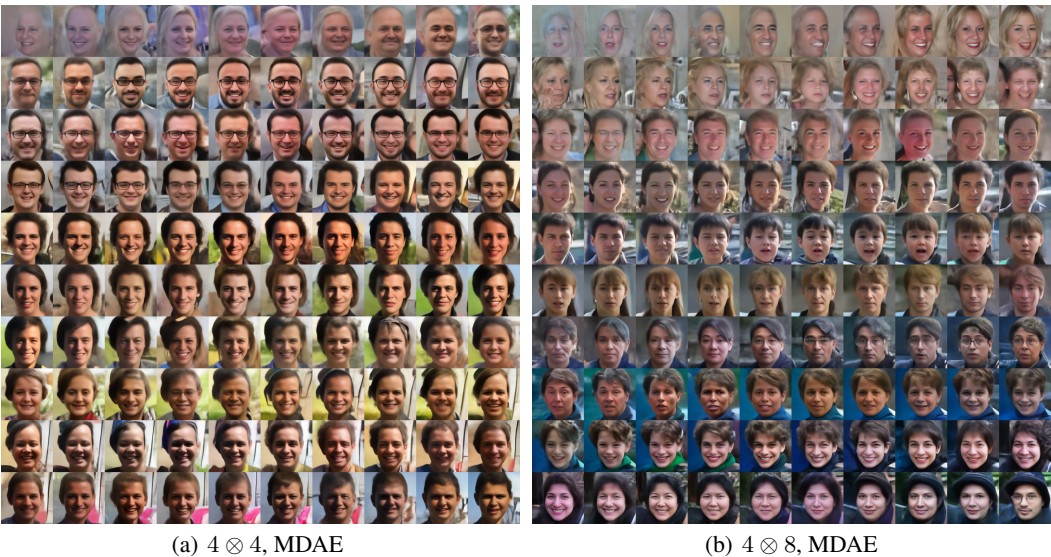

(a) $4 \otimes 4$, MDAE

(b) $4 \otimes 8$, MDAE

Figure 3: (*WJS chains on FFHQ-256*) The chains are shown skipping $\Delta k = 10$ steps (*no warmup*). We used Algorithm 1 with $\delta = 2, \gamma = 1/2, u = 1$. Transitions are best viewed electronically.

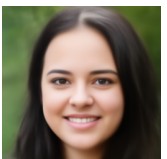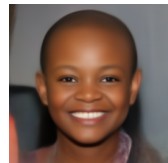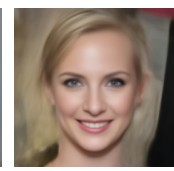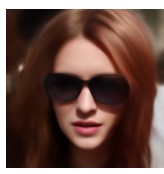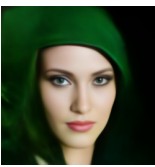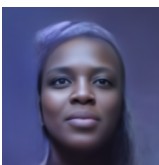

Figure 4: ($4 \otimes 8$ *gallery*) Samples from our FFHQ-256, MDAE, $4 \otimes 8$ model.

## 6   EXPERIMENTAL RESULTS

For all models with the MDAE and MEM$^2$ parametrizations, we used U$^2$-Net (Qin et al., 2020) network architecture, a recent variant of UNet (Ronneberger et al., 2015), with a few simple changes (see Appendix E). For the MUVB parametrization for MNIST (evaluated in Sec. G.3), we used Residual networks for the encoder, decoder and metaencoder. Despite the inherent complexity of the task of learning high-dimensional energy models, it is notable that our framework results in a single least-squares objective function, which we optimized using Adam (Kingma & Ba, 2014) without additional learning rate schedules. Additional details of the experimental setup are in **Appendix E**.

Arguably, the ultimate test for an energy model is whether one can generate realistic samples from it with *a single fast mixing MCMC chain that explores all modes of the distribution indefinitely*, starting from random noise. We informally refer to them as **lifelong Markov chains**. To give a physical picture, a gas of molecules that has come in contact with a thermal reservoir does not stop mid air after thermalizing—arriving at its "first sample"—it continues generating samples from the Boltzmann distribution as long as the physical system exists. To meet this challenge, the energy landscape must not have any pathologies that cause the sampling chain to break or get stuck in certain modes. In addition, we need fast mixing (Langevin) MCMC algorithms, a very active area of research by itself.

To put this goal in context, in recent energy models for high-dimensional data (Xie et al., 2016; 2018; Nijkamp et al., 2019; Du & Mordatch, 2019; Zhao et al., 2020; Du et al., 2020; Xie et al., 2021), sampling using MCMC quickly breaks or collapses to a mode and chains longer than a few hundred steps were not reported. Thus, evaluation in prior work relies on samples from independent MCMC chains, in addition by using heuristics like "replay buffer" (Du & Mordatch, 2019). In this work, we report FID scores obtained by single MCMC chains, the first result of its kind, which we consider as a benchmark for future works on long run MCMC chains (see **Table 4** for numerical comparisons).

For MNIST, we obtain fast-mixing chains for over **1 M** steps using MDAE and MUVB. On CIFAR-10 and FFHQ-256, we obtain stable chains up to 1 M and 10 K steps, respectively, using MDAE. The results are remarkable since energy models—that learn a *single* energy/score function—have never been scaled to 256×256 resolution images. The closest results are by Nijkamp et al. (2019) and Du et al. (2020) on CelebA (128×128)—in particular, our results in Figs. 3, 4 can be compared to (Nijkamp et al., 2019, Figs. 1, 2). For CIFAR-10, we report the FID score of *43.95* for $1 \otimes 8$ model, which we note is achieved by a **single MCMC chain** (Fig. 13); the closest result on FID score obtained for long run MCMC is *78.12* by Nijkamp et al. (2022) which is *not* on single chains, but on several parallel chains (in that paper the "long run" chains are 2 K steps vs. 1 M steps in this work).

Our detailed experimental results are organized in appendices as follows. **Appendix F** is devoted to an introduction to Langevin MCMC with demonstrations using our FFHQ-256 $4 \otimes 8$ model. **Appendix G** is devoted to MNIST experiments. At first we compare $1/4 \otimes 1$ and $1 \otimes 16$ (Sec. G.1). These models are statistically identical regarding the plug-in estimators (Eq. 10) and they arrive at similar training losses, but very different as generative models. This sheds light on the question why we considered such high noise levels in designing experiments (Fig. 5). We then present the effect of increasing $M$ for a fixed $\sigma$, the issue of time complexity, and mixing time vs. image quality trade-off (Sec. G.2). The discussions in Sec. G.1 and Sec. G.2 are closely related. We then compare MDAE and MUVB for $1 \otimes 4$ with the message that MDAE generates sharper samples while MUVB has better mixing properties (Sec. G.3). We then present one example of a lifelong Markov chain (Sec. G.4). Our MDAE model struggles on the CIFAR-10 challenge due to optimization problems. The results and further discussion are presented in **Appendix H**. Finally, in **Appendix I** we show chains obtained for FFHQ-256, including the ones that some of the images in Fig. 4 were taken from.

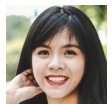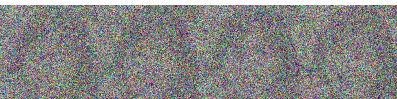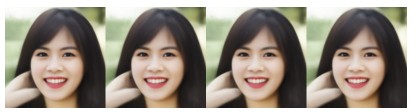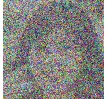

Figure 5: (*single-step multidenoising*) A clean data ($x$), a sample from MNM (**y**) for the $4 \otimes 4$ model (Fig. 3a), the outputs of MDAE ($\boldsymbol{\nu}$), and $\widehat{x}_{\mathrm{mean}}(\mathbf{y})$ are visualized from left to right. The outputs $\boldsymbol{\nu}_\theta(\mathbf{y})$ correspond to the Bayes estimator(s) which as expected from theory (Eq. 8) are indistinguishable.

## 7 RELATED WORK

Despite being a fundamental framework for denoising, empirical Bayes has been surprisingly absent in the DAE/DSM literature. Raphan & Simoncelli (2011) first mentioned the connection between empirical Bayes and score matching, but this work is practically unknown in the literature; for much of its development, DSM was all tied to DAEs (Vincent, 2011; Alain & Bengio, 2014); see (Bengio et al., 2013) for an extensive survey. Abstracting DSM away from DAE is due to Saremi et al. (2018) where they directly parametrized the energy function. In this work, we closed the circle, *from empirical Bayes to DAEs*. One can indeed use the MDAE objective and learn a generative model: *a highly generalized DAE* naturally arise from empirical Bayes. The important subtlety here is, as we remarked in Sec. 3.2, we can only arrive at this connection starting from our generalization of empirical Bayes, not the other way around. Of course, this core multimeasurement aspect of our approach, without which we do not have a generative model, does not exist in the DAE literature.[3]

To address the problem of choosing a noise level in DSM (Saremi et al., 2018), Song & Ermon (2019) studied it with multiple noise levels by summing up the losses using a weighing scheme. See (Li et al., 2019; Chen et al., 2020; Song & Ermon, 2020; Kadkhodaie & Simoncelli, 2020; Jolicoeur-Martineau et al., 2020) in that direction. The learning objectives in these models are based on heuristics in how different noise levels are weighted. Denoising diffusion models (Ho et al., 2020; Song et al., 2020; Gao et al., 2020) follow the same philosophy while being theoretically sound. Sampling in these models are based on annealing or reversing a diffusion process. *Our philosophy here is fundamentally different.* Even "noise levels" have very different meaning here, associated with the M noise channels of the factorial kernel. All noise levels (which we took to be *equal* in later parts, a meaningless choice in other methods) are encapsulated in a single energy/score function. Using Langevin MCMC we only sample highly noisy data **y**. Clean data in $\mathbb{R}^d$ is generated via a single step by computing $\widehat{x}(\mathbf{y})$.

## 8 CONCLUSION

We approached the problem of generative modeling by mapping a complex density in $\mathbb{R}^d$ to a smoother "dual" density, named M-density, in $\mathbb{R}^{Md}$. Permutation invariance is a design choice which we found particularly appealing. Using factorial kernels for density estimation and generative modeling have never been explored before and this work should open up many new avenues. We believe the topic of parametrization will play an especially important role in future developments. There is a unity in the parametrization schemes we proposed and studied in the paper, but much remains to be explored in understanding and extending the relationship between MEMs and MSMs.

Our MDAE parametrization scheme (an instance of MSM) is especially appealing for its simplicity and connections to the past literature. DAEs have a very rich history in the field of representation learning. But research on them as *generative models* stopped around 2015. Our hypothesis is that one must use very large noise to make them work as generative models for complex datasets in high dimensions. Our permutation invariant multimeasurement approach is an elegant solution since it allows for that choice at the cost of computation (large $M$), with only one parameter left to tune: $\sigma$! Crucially, the probabilistic interpretation is ingrained here, given by the algebraic relations between the MDAE output $\boldsymbol{\nu}(\mathbf{y})$, the Bayes estimator $\widehat{x}(\mathbf{y})$, and the score function $\nabla \log p(\mathbf{y})$.

---

[3]There are similarities between Eq. 5 in (Alain & Bengio, 2014) and Eq. 18 here. However, one is on barely noisy data ($\sigma \to 0$) in $\mathbb{R}^d$, the other on *multimeasurements* in $\mathbb{R}^{Md}$ for any (high) noise level $\sigma$. There, they arrived at Eq. 5 (with some effort) starting from a DAE objective. Here, we start with Eq. 18 (it *defines* the score function) and arrive at MDAE objective (Eq. 19), in one line of algebra.

## ACKNOWLEDGEMENT

We would like to thank Aapo Hyvärinen, Francis Bach, and Faustino Gomez for their valuable comments on the manuscript, and Vojtech Micka for help in running experiments.

## ETHICAL CONSIDERATIONS

By their very nature, generative models assign a high (implicit or explicit) likelihood to points in their training set. When samples are generated from a trained generative model, it is quite possible that some samples are very similar or identical to certain data points in the training set. This is important to keep in mind when the data used to train a generative model is confidential or contains personally identifiable information such as pictures of faces in the FFHQ-256 dataset (Karras et al., 2019). Care should be taken to abide by license terms and ethical principles when using samples from generative models. For FFHQ-256 in particular, please see terms of use at `https://github.com/NVlabs/ffhq-dataset/blob/master/README.md`. Additionally, we recommend considering the in-depth discussion on this subject by Prabhu & Birhane (2020) before deploying applications based on generative models.

## REPRODUCIBILITY STATEMENT

All important details of the experimental setup are provided in Appendix E. Our code is publicly available at `https://github.com/nnaisense/mems`.

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

## A  GAUSSIAN M-DENSITIES

We study the Bayes estimator for Gaussian MNMs analytically for $X = N(\mu, \sigma_0^2 I_d)$, where $\mu \in \mathbb{R}^d$. As in the example for Poisson MNM in the paper, the first step is to derive an expression for the joint density

$$p(\mathbf{y}) = \int \prod_m p_m(y_m|x)p(x)dx. \tag{24}$$

We start with $M = 2$. Next we perform the integral above:

$$\begin{aligned}
\log p(y_1, y_2) = &- \frac{\left\|y_1\right\|_2^2(\sigma_0^2 + \sigma_2^2) + \left\|y_2\right\|_2^2(\sigma_0^2 + \sigma_1^2) + \left\|\mu\right\|_2^2(\sigma_1^2 + \sigma_2^2)}{2(\sigma_1^2\sigma_2^2 + \sigma_2^2\sigma_0^2 + \sigma_0^2\sigma_1^2)} \\
&+ \frac{2\langle y_1, y_2\rangle\sigma_0^2 + 2\langle \mu, y_2\rangle\sigma_1^2 + 2\langle \mu, y_1\rangle\sigma_2^2}{2(\sigma_1^2\sigma_2^2 + \sigma_2^2\sigma_0^2 + \sigma_0^2\sigma_1^2)} - \log Z(\sigma_0, \sigma_1, \sigma_2),
\end{aligned} \tag{25}$$

where $Z$ is the partition function:

$$Z(\sigma_0, \sigma_1, \sigma_2) = \left((2\pi)^2(\sigma_0\sigma_1\sigma_2)^2(\sigma_0^{-2} + \sigma_1^{-2} + \sigma_2^{-2})\right)^{d/2}.$$

The multimeasurement Bayes estimator of $X$ is computed next via Eq. 7:

$$\widehat{x}(y_1, y_2) = \frac{\mu\sigma_1^2\sigma_2^2 + y_1\sigma_2^2\sigma_0^2 + y_2\sigma_0^2\sigma_1^2}{\sigma_0^2\sigma_1^2 + \sigma_0^2\sigma_2^2 + \sigma_1^2\sigma_2^2}. \tag{26}$$

As a sanity check, $\widehat{x}(\mathbf{y})$ is the same whether Eq. 7 is computed using $m = 1$ or $m = 2$. Another sanity check is the limit $\sigma_1 \to 0$, which we recover $\widehat{x}(y_1, y_2) = y_1 = x$. The *linear* relation (linear in $\mathbf{y}$ for $\mu = 0$) observed in Eq. 26 is a generalization of the well known results for $M = 1$, known as *Wiener filter* (Wiener, 1964).

A visualization of this result for a mixture of Gaussian is given in Fig. 1. Next we state the results for general $M$, and after that we derive the expression for the Bayes estimator $\widehat{x}(\mathbf{y})$.

It is convenient to define the following notations:

$$\Sigma_M := \prod_{m=0}^{M} \sigma_m^2 \cdot Z_M, \text{ where } Z_M := \prod_{m=0}^{M} \sigma_m^2 \cdot \sum_{m=0}^{M} \sigma_m^{-2},$$

$$\tilde{y} := \sum_{m=0}^{M} y_{M\setminus m}, \text{ where } y_{M\setminus m} := y_m \prod_{l \in [M]\setminus m} \sigma_l^2, \text{ and } y_0 := \mu.$$

With these notations, the expression for the M-density is given by:

$$\log p(\mathbf{y}) = - \sum_{m=0}^{M} \frac{\left\|y_m\right\|_2^2}{2\sigma_m^2} + \frac{\left\|\tilde{y}\right\|_2^2}{2\Sigma_M} - d\log\left((2\pi)^M \sqrt{Z_M}\right). \tag{27}$$

We can now derive $\widehat{x}(\mathbf{y})$ using Eq. 7. Given the expression above, the algebra is straightforward:

$$\begin{aligned}
\widehat{x}(\mathbf{y}) &= y_m + \sigma_m^2 \nabla_m \log p(\mathbf{y}) \\
&= y_m - y_m + \sigma_m^2 (2\Sigma_M)^{-1} \nabla_m \left\|\tilde{y}\right\|_2^2 \\
&= (2\prod_{l=0}^{M} \sigma_l^2 \cdot Z_M)^{-1} 2\sigma_m^2 \left(\prod_{l \in [M]\setminus m} \sigma_l^2\right) \tilde{y} \\
&= \tilde{y}/Z_M
\end{aligned} \tag{28}$$

*Note that the final expression is the same by carrying the algebra for any measurement index $m$.*

## B   M-DENSITY SCORE MATCHING ≡ MULTIMEASUREMENT DSM

Here we provide the score matching formalism (Hyvärinen, 2005) on M-densities and we arrive at a *multimeasurement denoising score matching* (MDSM) learning objective. The derivation closely follows (Vincent, 2011) using our notation. In what follows equalities are modulo additive constants (not functions of parameters $\theta$) which we denote by the color gray:

$$
\begin{aligned}
\mathcal{J}(\theta) &= \mathbb{E}_{\mathbf{y}}\big\| -\nabla_{\mathbf{y}}f_\theta(\mathbf{y}) - \nabla_{\mathbf{y}}\log p(\mathbf{y})\big\|_2^2 \\
&= \mathbb{E}_{\mathbf{y}}\big\|\nabla_{\mathbf{y}}f_\theta(\mathbf{y})\big\|_2^2 + 2\cdot\mathbb{E}_{\mathbf{y}}\langle\nabla_{\mathbf{y}}f_\theta(\mathbf{y}),\nabla_{\mathbf{y}}\log p(\mathbf{y})\rangle + \mathbb{E}_{\mathbf{y}}\big\|\nabla_{\mathbf{y}}\log p(\mathbf{y})\big\|_2^2 \\
&= \mathbb{E}_{\mathbf{y}}\big\|\nabla_{\mathbf{y}}f_\theta(\mathbf{y})\big\|_2^2 + 2\int p(\mathbf{y})\langle\nabla_{\mathbf{y}}f_\theta(\mathbf{y}),\nabla_{\mathbf{y}}\log p(\mathbf{y})\rangle d\mathbf{y} \\
&= \mathbb{E}_{\mathbf{y}}\big\|\nabla_{\mathbf{y}}f_\theta(\mathbf{y})\big\|_2^2 + 2\int \langle\nabla_{\mathbf{y}}f_\theta(\mathbf{y}),\nabla_{\mathbf{y}}p(\mathbf{y})\rangle d\mathbf{y} \\
&= \mathbb{E}_{\mathbf{y}}\big\|\nabla_{\mathbf{y}}f_\theta(\mathbf{y})\big\|_2^2 + 2\int \langle\nabla_{\mathbf{y}}f_\theta(\mathbf{y}),\nabla_{\mathbf{y}}p(\mathbf{y}|x)\rangle p(x)dxd\mathbf{y} \\
&= \mathbb{E}_{\mathbf{y}}\big\|\nabla_{\mathbf{y}}f_\theta(\mathbf{y})\big\|_2^2 + 2\int \langle\nabla_{\mathbf{y}}f_\theta(\mathbf{y}),p(\mathbf{y}|x)^{-1}\nabla_{\mathbf{y}}p(\mathbf{y}|x)\rangle p(\mathbf{y}|x)p(x)dxd\mathbf{y} \\
&= \mathbb{E}_{\mathbf{y}}\big\|\nabla_{\mathbf{y}}f_\theta(\mathbf{y})\big\|_2^2 + 2\cdot\mathbb{E}_{(x,\mathbf{y})}\langle\nabla_{\mathbf{y}}f_\theta(\mathbf{y}),\nabla_{\mathbf{y}}\log p(\mathbf{y}|x)\rangle \\
&= \mathbb{E}_{(x,\mathbf{y})}\big\|\nabla_{\mathbf{y}}f_\theta(\mathbf{y})\big\|_2^2 + 2\cdot\mathbb{E}_{(x,\mathbf{y})}\langle\nabla_{\mathbf{y}}f_\theta(\mathbf{y}),\nabla_{\mathbf{y}}\log p(\mathbf{y}|x)\rangle + \mathbb{E}_{(x,\mathbf{y})}\big\|\nabla_{\mathbf{y}}\log p(\mathbf{y}|x)\big\|_2^2 \\
&= \mathbb{E}_{(x,\mathbf{y})}\big\| -\nabla_{\mathbf{y}}f_\theta(\mathbf{y}) - \nabla_{\mathbf{y}}\log p(\mathbf{y}|x)\big\|_2^2
\end{aligned}
\tag{29}
$$

The M-density score matching learning objective is given by the first identity above and the last equality is the MDSM objective:

$$
\mathcal{J}(\theta) = \mathbb{E}_{(x,\mathbf{y})}\big\| -\nabla_{\mathbf{y}}f_\theta(\mathbf{y}) - \nabla_{\mathbf{y}}\log p(\mathbf{y}|x)\big\|_2^2
\tag{30}
$$

For Gaussian MNMs, the MDSM learning objective simplifies to:

$$
\mathcal{J}(\theta) = \sum_{m=1}^{M}\mathcal{J}_m(\theta),\ \text{where}\ \mathcal{J}_m = \mathbb{E}_{(x,\mathbf{y})}\Big\| -\nabla_m f_\theta(\mathbf{y}) + \frac{y_m - x}{\sigma_m^2}\Big\|_2^2.
\tag{31}
$$

## C   MUVB: AN UNNORMALIZED LATENT VARIABLE MODEL

In this section we give a formulation of the energy function parametrization scheme which is named *multimeasurement unnormalized variational Bayes* (MUVB). Conceptually, MUVB is motivated by the goal of connecting empirical Bayes' *denoising* framework with variational Bayes' *inference* machinery. There seem to be fundamental challenges here since one model is based on least-squares estimation, the other based on the principle of maximum likelihood. In what follows, this "fundamental conflict" takes the shape of latent variable model that is *unnormalized*. There have been recent sudies along these lines (Saremi, 2020a;b) that we expand on; in particular we introduce the novel *metaencoder*. At a high level, the idea here is to set up a latent variable model for M-densities and use the variational free energy as the energy function. We refer to (Jordan et al., 1999) for an introduction to variational methods. In what follows, we use the lighter notation for $\sigma \otimes M$ models (Remark 2) that we also used in Sec. 4.

In its simplest form, the latent variable model for M-densities is set up by the following choices:

- The prior over latent variables which we take to be Gaussian

$$
p(z) = \mathcal{N}(z; 0, I_{d_z}).
$$

- The approximate posterior over latent variables $q_\phi(z|\mathbf{y})$ which is taken to be the factorized Gaussian, a standard choice in the literature.

- For the conditional density $p_\eta(\mathbf{y}|z)$ we can consider the following

$$
p_\eta(\mathbf{y}|z) = \mathcal{N}(\mathbf{y}; \boldsymbol{\nu}_\eta(z), \sigma^2 I_{Md}),
$$

which may seem as a especially "natural" choice since samples from the M-density are obtained by adding multimeasurement noise to clean samples.

In the variational autoencoder parlance, $\phi$ are the *encoder*'s parameters, $\eta$ the *decoder*'s parameters, and $\boldsymbol{\nu} = (\nu_1, \ldots, \nu_M)$ the decoder's outputs. With this setup, the variational free energy is easily derived which we take to be the energy function as follows ($\theta = (\phi, \eta)$):

$$f_\theta(\mathbf{y}) = \frac{1}{2\sigma^2} \mathbb{E}_{q_\phi(z|\mathbf{y})} \|\mathbf{y} - \boldsymbol{\nu}_\eta(z)\|_2^2 + \text{KL}(q_\phi(z|\mathbf{y}) \| p(z)),$$

where the KL divergence is derived in closed form, and the expectation is computed by sampling $z \sim q_\phi(z|\mathbf{y})$ via the reparametrization trick (Kingma & Welling, 2013).

In this machinery, in the inference step we are forced to have normalized (and tractable) posteriors since we must take samples from $q_\phi(z|\mathbf{y})$. What about the conditional density $p(\mathbf{y}|z)$? If we were to formalize a VAE for modeling M-densities we had to have a *normalized* conditional density $p(\mathbf{y}|z)$ since that framework is based on the principle of maximum *likelihood*. What is intriguing about our setup is that since our goal is to learn the unnormalized $p(\mathbf{y})$, we can consider conditional densities $p(\mathbf{y}|z)$ that are *unnormalized*. An intuitive choice is the following that we name *metalikelihood*:

$$p_{(\eta,\zeta)}(\mathbf{y}|z) \propto \exp\left(-\frac{1}{2\sigma^2}\|\mathbf{y} - \boldsymbol{\nu}_\eta(z)\|_2^2 - h_\zeta(\mathbf{y}, \boldsymbol{\nu}_\eta(z))\right).$$

Now, the energy function takes the following form ($\theta = (\phi, \eta, \zeta)$):

$$f_\theta(\mathbf{y}) = \mathbb{E}_{q_\phi(z|\mathbf{y})}\left(\frac{1}{2\sigma^2}\|\mathbf{y} - \boldsymbol{\nu}_\eta(z)\|_2^2 + h_\zeta(\mathbf{y}, \boldsymbol{\nu}_\eta(z))\right) + \text{KL}(q_\phi(z|\mathbf{y}) \| p(z)). \tag{32}$$

This is the final form of MUVB energy function. We refer to $h_\zeta$ by metaencoder. Its input is $(\mathbf{y}, \nu(z; \eta))$ (note that $z$ itself is a function of $\mathbf{y}$ via inference), and one can simply use any encoder architecture to parametrize it. In our implementation, we used the standard VAE pipeline and we reused the encoder architecture to parametrize the metaencoder.

## D  TWO WALK-JUMP SAMPLING ALGORITHMS

In this section we provide the walk-jump sampling (WJS) algorithm based on different discretizations of underdamped Langevin diffusion (Eq. 23). The first is due to Sachs et al. (2017) which has also been used by Arbel et al. (2020) for generative modeling. The second is based on a recent algorithm analyzed by Cheng et al. (2018) which we implemented and give the detailed algorithm here. In addition, we extended their calculation to general friction (they set $\gamma = 2$ in the paper) and provide an extension of the (Cheng et al., 2018, Lemma 11) for completeness (the proof closely follows the calculation in the reference). In both cases we provide the algorithm for $\sigma \otimes M$ models.

### D.1  WALK-JUMP SAMPLING ALGORITHM I

---

**Algorithm 1:** Walk-jump sampling (Saremi & Hyvärinen, 2019) using the discretization of Langevin diffusion by Sachs et al. (2017). The for loop corresponds to the dashed arrows (*walk*) in Fig. 2 and line 14 ($\langle \cdot \rangle$ is computed over the measurement indices $m$) to the solid arrow (*jump*).

---

1:  **Input** $\delta$ (step size), $u$ (inverse mass), $\gamma$ (friction), $K$ (steps taken)
2:  **Input** Learned energy function $f(\mathbf{y})$ or score function $\mathbf{g}(\mathbf{y}) \approx \nabla \log p(\mathbf{y})$
3:  **Ouput** $\widehat{X}_K$
4:  $\mathbf{Y}_0 \sim \text{Unif}([0,1]^{Md})$
5:  $\mathbf{V}_0 \leftarrow 0$
6:  **for** $k = [0, \ldots, K)$ **do**
7:    $\mathbf{Y}_{k+1} \leftarrow \mathbf{Y}_k + \delta \mathbf{V}_k/2$
8:    $\boldsymbol{\Psi}_{k+1} \leftarrow -\nabla_\mathbf{y} f(\mathbf{Y}_{k+1})$ or $\boldsymbol{\Psi}_{k+1} \leftarrow \mathbf{g}(\mathbf{Y}_{k+1})$
9:    $\mathbf{V}_{k+1} \leftarrow \mathbf{V}_k + u\delta\boldsymbol{\Psi}_{k+1}/2$
10:   $\mathbf{B}_{k+1} \sim N(0, I_{Md})$
11:   $\mathbf{V}_{k+1} \leftarrow \exp(-\gamma\delta)\mathbf{V}_{k+1} + u\delta\boldsymbol{\Psi}_{k+1}/2 + \sqrt{u(1 - \exp(-2\gamma\delta))}\mathbf{B}_{k+1}$
12:   $\mathbf{Y}_{k+1} \leftarrow \mathbf{Y}_{k+1} + \delta\mathbf{V}_{k+1}/2$
13:  **end for**
14:  $\widehat{X}_K \leftarrow \langle Y_{K,m} - \sigma^2 \nabla_m f(\mathbf{Y}_K)\rangle$ or $\widehat{X}_K \leftarrow \langle Y_{K,m} + \sigma^2 g_m(\mathbf{Y}_K)\rangle$

---

## D.2 Walk-Jump Sampling Algorithm II

Before stating the algorithm, we extend the calculation in Cheng et al. (2018, Lemma 11) to general friction, which we used in our implementation of their Langevin MCMC algorithm. The algorithm is stated following the proof of the lemma which closely follows Cheng et al. (2018).

The starting point is solving the diffusion Eq. 23. With the initial condition $(\mathbf{y}_0, \mathbf{v}_0)$, the solution $(\mathbf{y}_t, \mathbf{v}_t)$ to the discrete underdamped Langevin diffusion is ($t$ is considered small here)

$$
\mathbf{v}_t = \mathbf{v}_0 e^{-\gamma t} - u \left( \int_0^t e^{-\gamma(t-s)} \nabla f(\mathbf{y}_0) ds \right) + \sqrt{2\gamma u} \int_0^t e^{-\gamma(t-s)} d\mathbf{B}_s,
$$

$$
\mathbf{y}_t = \mathbf{y}_0 + \int_0^t \mathbf{v}_s ds,
$$

(33)

These equations above are then used in the proof of the following lemma.

**Lemma 1.** *Conditioned on* $(\mathbf{y}_0, \mathbf{v}_0)$*, the solution of underdamped Langevin diffusion (Eq. 23) integrated up to time* $t$ *is a Gaussian with conditional means*

$$
\mathbb{E}[\mathbf{v}_t] = \mathbf{v}_0 e^{-\gamma t} - \gamma^{-1} u (1 - e^{-\gamma t}) \nabla f(\mathbf{y}_0)
$$

$$
\mathbb{E}[\mathbf{y}_t] = \mathbf{y}_0 + \gamma^{-1}(1 - e^{-\gamma t}) \mathbf{v}_0 - \gamma^{-1} u \left( t - \gamma^{-1}(1 - e^{-\gamma t}) \right) \nabla f(\mathbf{y}_0),
$$

(34)

*and with conditional covariances*

$$
\mathbb{E}[(\mathbf{y}_t - \mathbb{E}[\mathbf{y}_t])(\mathbf{y}_t - \mathbb{E}[\mathbf{y}_t])^\top] = \gamma^{-1} u \left( 2t - 4\gamma^{-1}(1 - e^{-\gamma t}) + \gamma^{-1}(1 - e^{-2\gamma t}) \right) \cdot I_{Md},
$$

$$
\mathbb{E}[(\mathbf{v}_t - \mathbb{E}[\mathbf{v}_t])(\mathbf{v}_t - \mathbb{E}[\mathbf{v}_t])^\top] = u(1 - e^{-2\gamma t}) \cdot I_{Md},
$$

(35)

$$
\mathbb{E}[(\mathbf{y}_t - \mathbb{E}[\mathbf{y}_t])(\mathbf{v}_t - \mathbb{E}[\mathbf{v}_t])^\top] = \gamma^{-1} u (1 - 2e^{-\gamma t} + e^{-2\gamma t}) \cdot I_{Md}.
$$

*Proof.* Computation of conditional means is straightforward as the Brownian motion has zero means:

$$
\mathbb{E}[\mathbf{v}_t] = \mathbf{v}_0 e^{-\gamma t} - \gamma^{-1} u (1 - e^{-\gamma t}) \nabla f(\mathbf{y}_0)
$$

$$
\mathbb{E}[\mathbf{y}_t] = \mathbf{y}_0 + \gamma^{-1}(1 - e^{-\gamma t}) \mathbf{v}_0 - \gamma^{-1} u \left( t - \gamma^{-1}(1 - e^{-\gamma t}) \right) \nabla f(\mathbf{y}_0),
$$

All the conditional covariances only involve the Brownian motion terms. We start with the simplest:

$$
\mathbb{E}[(\mathbf{v}_t - \mathbb{E}[\mathbf{v}_t])(\mathbf{v}_t - \mathbb{E}[\mathbf{v}_t])^\top] = 2\gamma u \, \mathbb{E} \left[ \left( \int_0^t e^{-\gamma(t-s)} d\mathbf{B}_s \right) \left( \int_0^t e^{-\gamma(t-s)} d\mathbf{B}_s \right)^\top \right]
$$

$$
= 2\gamma u \left( \int_0^t e^{-2\gamma(t-s)} ds \right) \cdot I_{Md}
$$

$$
= u(1 - e^{-2\gamma t}) \cdot I_{Md}
$$

(36)

From Eq. 33, the Brownian motion term for $\mathbf{y}_t$ is given by

$$
\sqrt{2\gamma u} \int_0^t \left( \int_0^r e^{-\gamma(t-s)} d\mathbf{B}_s \right) dr = \sqrt{2\gamma u} \int_0^t e^{\gamma s} \left( \int_s^t e^{-\gamma r} dr \right) d\mathbf{B}_s
$$

$$
= \sqrt{2\gamma^{-1} u} \int_0^t \left( 1 - e^{-\gamma(t-s)} \right) d\mathbf{B}_s.
$$

The conditional covariance for $\mathbf{y}_t$ follows similar to Eq. 36:

$$
\mathbb{E}[(\mathbf{y}_t - \mathbb{E}[\mathbf{y}_t])(\mathbf{y}_t - \mathbb{E}[\mathbf{y}_t])^\top] = 2\gamma^{-1} u \left( \int_0^t \left( 1 - e^{-\gamma(t-s)} \right)^2 ds \right) \cdot I_{Md}
$$

$$
= \gamma^{-1} u \left( 2t - 4\gamma^{-1}(1 - e^{-\gamma t}) + \gamma^{-1}(1 - e^{-2\gamma t}) \right) \cdot I_{Md}
$$

(37)

Finally the conditional covariance between $\mathbf{y}_t$ and $\mathbf{v}_t$ is given by

$$
\mathbb{E}[(\mathbf{y}_t - \mathbb{E}[\mathbf{y}_t])(\mathbf{v}_t - \mathbb{E}[\mathbf{v}_t])^\top] = 2u \, \mathbb{E} \left[ \left( \int_0^t \left( 1 - e^{-\gamma(t-s)} \right) d\mathbf{B}_s \right) \left( \int_0^t e^{-\gamma(t-s)} d\mathbf{B}_s \right)^\top \right]
$$

$$
= 2u \left( \int_0^t \left( 1 - e^{-\gamma(t-s)} \right) e^{-\gamma(t-s)} ds \right) \cdot I_{Md}
$$

$$
= \gamma^{-1} u (1 - 2e^{-\gamma t} + e^{-2\gamma t}) \cdot I_{Md}
$$

$\square$

---

**Algorithm 2:** Walk-Jump Sampling (WJS) using the discretization of Langevin diffusion from Lemma 1. Below, we provide the Cholesky decomposition of the conditional covariance matrix.

---

1: **Input** $\delta$ (step size), $u$ (inverse mass), $\gamma$ (friction), $K$ (steps taken)
2: **Input** Learned energy function $f(\mathbf{y})$ or score function $\mathbf{g}(\mathbf{y}) \approx \nabla \log p(\mathbf{y})$
3: **Ouput** $\widehat{X}_K$
4: $\mathbf{Y}_0 \sim \text{Unif}([0,1]^{Md})$
5: $\mathbf{V}_0 \leftarrow 0$
6: **for** $k = [0, \ldots, K)$ **do**
7: $\quad \Psi_k \leftarrow -\nabla_{\mathbf{y}} f(\mathbf{Y}_k)$ or $\Psi_k \leftarrow \mathbf{g}(\mathbf{Y}_k)$
8: $\quad \mathbf{V}_{k+1} \leftarrow \mathbf{V}_k e^{-\gamma\delta} + \gamma^{-1} u(1 - e^{-\gamma\delta})\Psi_k$
9: $\quad \mathbf{Y}_{k+1} \leftarrow \mathbf{Y}_k + \gamma^{-1}(1 - e^{-\gamma\delta})\mathbf{V}_k + \gamma^{-1} u\left(\delta - \gamma^{-1}(1 - e^{-\gamma\delta})\right)\Psi_k$
10: $\quad \mathbf{B}_{k+1} \sim N(0, I_{2Md})$
11: $\quad \begin{pmatrix} \mathbf{Y}_{k+1} \\ \mathbf{V}_{k+1} \end{pmatrix} \leftarrow \begin{pmatrix} \mathbf{Y}_{k+1} \\ \mathbf{V}_{k+1} \end{pmatrix} + \mathbf{L}\mathbf{B}_{k+1}$ $\quad$ // see Eq. 38
12: **end for**
13: $\widehat{X}_K \leftarrow \langle Y_{K,m} - \sigma^2 \nabla_m f(\mathbf{Y}_K)\rangle$ or $\widehat{X}_K \leftarrow \langle Y_{K,m} + \sigma^2 g_m(\mathbf{Y}_K)\rangle$

---

The Cholesky decomposition $\mathbf{\Sigma} = \mathbf{L}\mathbf{L}^\top$ for the conditional covariance in Lemma 1 is given by:

$$\mathbf{L} = \begin{pmatrix} \Sigma_{\mathbf{yy}}^{1/2} \cdot I_{Md} & 0 \\ \Sigma_{\mathbf{yy}}^{-1/2} \Sigma_{\mathbf{yv}} \cdot I_{Md} & (\Sigma_{\mathbf{vv}} - \Sigma_{\mathbf{yv}}^2/\Sigma_{\mathbf{yy}})^{1/2} \cdot I_{Md} \end{pmatrix}, \tag{38}$$

where

$$\Sigma_{\mathbf{yy}} = \gamma^{-1} u \left(2\delta - 4\gamma^{-1}(1 - e^{-\gamma\delta}) + \gamma^{-1}(1 - e^{-2\gamma\delta})\right) \tag{39}$$

$$\Sigma_{\mathbf{vv}} = u(1 - e^{-2\gamma\delta}) \tag{40}$$

$$\Sigma_{\mathbf{yv}} = \gamma^{-1} u(1 - 2e^{-\gamma\delta} + e^{-2\gamma\delta}) \tag{41}$$

**Algorithm 1 vs. Algorithm 2** The two algorithms presented in this section have very different properties in our high dimensional experiments. In general we found Algorithm 1 (A1) to behave similarly to overdamped version (see Appendix F for some illustrations) albeit with much faster mixing. We found friction $\gamma$ to play a very different role in comparing the two algorithms, also apparent in comparing the mathematical expressions in both algorithms: in A1, the friction *only* appears in the form $\exp(-\gamma\delta)$; it is more complex in A2. Similar discrepancy is found regarding $u$.

**Remark 4** (Initialization scheme). *The initialization $\mathbf{Y}_0 \sim Unif([0,1]^{Md})$ used in the algorithms is the conventional choice, but for very large $\sigma$ we found it effective to initialize the chain by further adding the Gaussian noise in $\mathbb{R}^{Md}$:*

$$\mathbf{Y}_0 = \boldsymbol{\epsilon}_0 + \boldsymbol{\varepsilon}_0, \text{ where } \boldsymbol{\epsilon}_0 \sim Unif([0,1]^{Md}), \boldsymbol{\varepsilon}_0 \sim N(0, \sigma^2 I_d)$$

*Otherwise, with large step sizes $\delta = O(\sigma)$ the chains starting with the uniform distribution could break early on. We found this scheme reliable, and it is well motivated since M-density $p(\mathbf{y})$ is obtained by convolving $p(x)$ with the Gaussian MNM. With this initialization one starts "relatively close" to the M-density manifold and this is more pronounced for larger $\sigma$. However, this initialization scheme is based on heuristics (as is the conventional one).*

**Remark 5** (The jump in WJS is asynchronous). *The jump in WJS is asynchronous (Fig. 2) and it can also be taken inside the for loop in the algorithms provided without affecting the Langevin chain.*

**Remark 6** (The $\Delta k$ notation). *In running long chains we set $K$ to be large, and we report the jumps (taken inside the for loops in the algorithms provided) with a certain fixed frequency denoted by $\Delta k$.*

**Remark 7** (WJS for Poisson M-densities). *WJS is a general sampling algorithm, schematized in Fig. 2. However, for Poisson MNMs it will involve sampling a discrete distribution in $\mathbb{N}^{Md}$ which we do not know how to do efficiently in high dimensions (note that Langevin MCMC cannot be used since the score function is not well defined for discrete distributions). We can use Metropolis-Hastings algorithm and its popular variant Gibbs sampling but these algorithms are very slow in high dimensions since at each iteration of the sampling algorithm in principle only 1 dimension out of Md shall be updated, so in the general case the mixing time is at best of $O(Md)$; one can use "blocking" techniques but they are problem dependent and typically complex (Andrieu et al., 2003).*

# E EXPERIMENTAL SETUP

**Datasets**  Our experiments were conducted on the MNIST (LeCun et al., 1998), CIFAR-10 (Krizhevsky, 2009) and FFHQ-256 (Karras et al., 2019; pre-processed version by Child, 2020) datasets. The CIFAR-10 and FFHQ-256 training sets were augmented with random horizontal flips during training.

**Network Architectures**  For all experiments with the MDAE and MEM$^2$ parametrization, the U$^2$Net network architecture (Qin et al., 2020) was chosen since it is a recent and effective variant of the UNet (Ronneberger et al., 2015) that has been successfully applied to various pixel-level prediction problems. However, we have not experimented with alternatives. Minor adjustments were made to the U$^2$Net from the original paper: we removed all normalization layers, and switched the activation function to $x \mapsto x \cdot \text{sigmoid}(x)$ (Elfwing et al., 2017; Ramachandran et al., 2017). To approximately control the variance of inputs to the network in each dimension, even if the noise level $\sigma$ is large, the noisy input values were divided by a scaling factor $\sqrt{0.225^2 + \sigma^2}$. We also adjusted the number of "stages" in the network, stage heights and layer widths (see Table 1 for details), in order to fit reasonable batch sizes on available GPU hardware and keep experiment durations limited to a few days (see Table 2).

For experiments with the MUVB parametrization (the MNIST model in Sec. G.3), we used Residual convolutional networks utilizing a bottleneck block design with skip connections and average pooling layers (Lin et al., 2013; Srivastava et al., 2015; He et al., 2016). The architecture of the encoder and metaencoder were kept the same, while the decoder used the encoder architecture in reverse, with the pooling layers substituted with nearest neighbour upsampling. See Table 3 for a detailed description of the encoder architecture.

**Remark 8.** *Strictly speaking, for $\sigma \otimes M$ models the energy function is permutation invariant and* **y** *should be treated as $\{y_1, \ldots, y_M\}$, a set consisting of $M$ Euclidean vectors. This is however difficult to enforce while keeping the model* expressive. *We note that in our model the permutation symmetry is a "fragile symmetry" which one can easily break by making $\sigma_m$ just slightly different from each other. Therefore, we designed the architecture to be used for the general setting of $\sigma_m$ different from each other, and in experiments we just set them to be equal.*

**Training**  All models were trained using the Adam (Kingma & Ba, 2014) optimizer with the default setting of $\beta_1 = 0.9, \beta_2 = 0.999$ and $\epsilon = 10^{-8}$. Table 2 lists the main hyperparameters used and hardware requirements for training MDAE models for each dataset. The resulting training curves are shown in Fig. 6, showing the stability of the optimization with a relatively simple setup.

**Software**  All models were implemented in the CPython (v3.8) library PyTorch v1.9.0 (Paszke et al., 2017) using the PyTorch Lightning framework v1.4.6 (Falcon et al.).

Table 1: U$^2$Net architecture used for MDAE and MEM$^2$ parametrizations for all datasets. All layer widths (number of channels) were expanded by a width factor for CIFAR-10 and FFHQ-256.

| Stage | Height | In channels | Mid channels | Out channels | Side |
|-------|--------|-------------|--------------|--------------|------|
| Encoder 1 | 6 | 3 or 1 | 64 | 64 | |
| Encoder 2 | 5 | 64 | 128 | 128 | |
| Encoder 3 | 4 | 128 | 128 | 256 | |
| Encoder 4 | 3 | 256 | 256 | 512 | |
| Encoder 5 | 3 | 512 | 256 | 512 | 512 |
| Decoder 4 | 3 | 1024 | 128 | 256 | 256 |
| Decoder 3 | 3 | 512 | 128 | 128 | 128 |
| Decoder 2 | 5 | 256 | 128 | 64 | 64 |
| Decoder 1 | 6 | 128 | 64 | 64 | 64 |

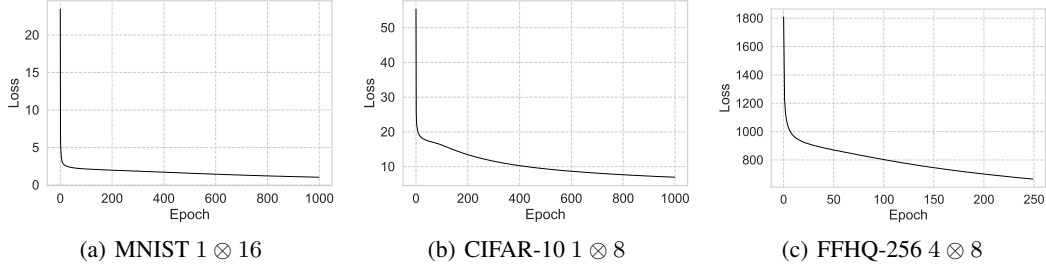

(a) MNIST $1 \otimes 16$    (b) CIFAR-10 $1 \otimes 8$    (c) FFHQ-256 $4 \otimes 8$

Figure 6: MDAE training plots demonstrate that optimization is stable for all datasets. Loss values on the y-axis in these plots represent the value of our objective function (Eq. 14).

Table 2: Main hyperparameters and computational resources used for training MDAE models.

|  | MNIST $1 \otimes 16$ | CIFAR-10 $1 \otimes 8$ | FFHQ-256 $4 \otimes 8$ |
|---|---|---|---|
| Width factor | 1 | 2 | 2 |
| Learning rate | 0.00002 | 0.00002 | 0.00002 |
| Total batch size | 128 | 256 | 16 |
| GPUs | 1×GTX Titan X | 4×GTX Titan X | 4×V100 |
| Run Time | ≈2.3 days (1000 epochs) | ≈2 days (1000 epochs) | ≈2.75 days (250 epochs) |

Table 3: MUVB Encoder architecture for MNIST. Each row indicates a sequence of transformations in the first column, and the spatial resolution of the resulting output in the second column. **Block{N}×{D}** denotes a sequence of D blocks, each consisting of four convolutional layers with a skip connection using a "bottleneck" design: the layers have widths $[N, 0.25 * N, 0.25 * N, N]$ and kernel sizes $[1, 3, 3, 1]$, except when the input resolution becomes smaller than 3, in which case all the kernel sizes are 1.

| Module | Output resolution |
|---|---|
| Input | 28×28 |
| Block128×2 | 28×28 |
| AveragePool | 14×14 |
| Block256×2 | 14×14 |
| AveragePool | 7×7 |
| Block512×2 | 7×7 |
| AveragePool | 4×4 |
| Block1024×2 | 4×4 |
| AveragePool | 2×2 |
| Block1024×2 | 2×2 |
| AveragePool | 1×1 |
| Block1024×1 | 1×1 |

## F  Langevin MCMC Laboratory

In this section, we give a light tutorial on Langevin MCMC used in the walk phase of WJS. Perhaps the most interesting result enabled by *multimeasurement generative models* is that Langevin MCMC in high dimensions becomes stable enough for us to examine and dissect the performance of various Langevin MCMC variants in practice.

The reader might ask why we used *underdamped* Langevin MCMC versus its well-known overdamped version. The reason is the recent theoretical developments discussed in Sec. 5 that show much faster *dimension* dependence for the convergence (mixing time), $O(\sqrt{d})$ vs. $O(d)$, for underdamped Langevin MCMC. To give an idea regarding the dimension, for the $4 \otimes 8$ model on FFHQ-256, $Md \approx 10^6$. Our trained model for $4 \otimes 8$ is used throughout this section. In addition, the MCMC is initialized with the fixed random seed throughout this section.

### F.1  Overdamped Langevin MCMC

We start with overdamped Langevin MCMC which is simpler and give us some intuitions on the behavior of Langevin MCMC in general. The algorithm is based on the following stochastic iterations (setting inverse mass $u = 1$):

$$\mathbf{y}_{k+1} \leftarrow \underbrace{\mathbf{y}_k - \frac{\delta^2}{2}\nabla f(\mathbf{y}_k)}_{A_k} + \underbrace{\delta \cdot \varepsilon_k}_{B_k}, \text{ where } \varepsilon_k \sim N(0, I_{Md}). \tag{42}$$

Comparing above with Eq. 7, it is intuitive to consider setting $\delta = O(\sigma)$. We start with $\delta = \sigma$: in this case $A_k$ pulls noisy data to clean manifold (note the important extra factor of $1/2$ however) and $B_k$ term corresponds to adding multimeasurement noise sampled from $N(0, \sigma^2 I_{Md})$. The overall effect is that one starts "walking" on the manifold of the M-density. And if that is the case, the jumps computed by $\widehat{x}(\mathbf{y}_k)$ should give samples close to the manifold of the (clean) data distribution—samples from $p(x)$. We start with that experiment by setting $\delta = \sigma$. Note that in walk-jump sampling we sample the M-density in $\mathbb{R}^{Md}$ generating highly noisy data for $\sigma = 4$; we only show clean (jump) samples in $\mathbb{R}^d$. This is shown next by skipping every 30 steps, i.e. $\Delta k = 30$:

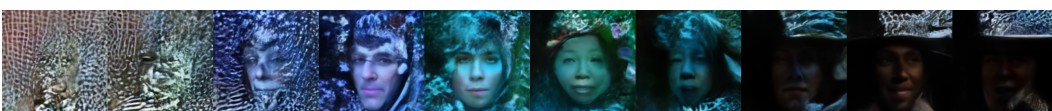

Our intuition was partly correct in that the sampler manages to "capture modes" but the step size is clearly too high. Now consider $\delta = \sigma/2$ and $\Delta k = 100$:

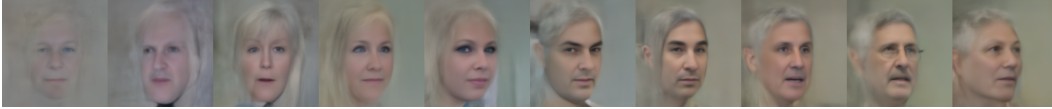

Continuing the chain, now shown for $\Delta k = 1000$, we arrive at

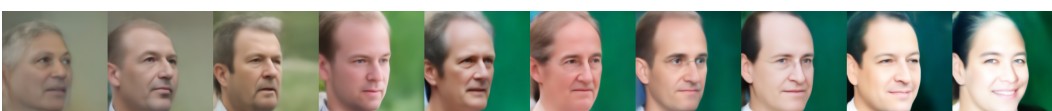

Theoretically, to converge to the true distribution (associated with the M-density) the step size should be small, where "small" is dictated by how "close" one wishes to be to the true distribution, e.g. see (Cheng et al., 2018, Theorem 1). As we see above, even for $\delta = 2$ the sampler does not converge after 1000 steps, but a fixed $\delta = \sigma/2$ appears to be too high for this model. By continuing with the chain with the step size $\delta = \sigma/2$ one starts to gradually move *away* from the manifold as shown next:

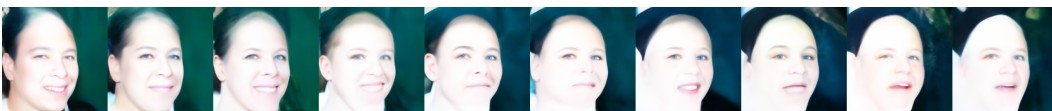

### F.2 Underdamped Langevin MCMC

In this section we focus on underdamped Langevin MCMC. The algorithms are based on discretizing underdamped Langevin diffusion (Eq. 23). In this paper, we considered two such discretization schemes, by Sachs et al. (2017) that we used in Algorithm 1, and the one due to Cheng et al. (2018) that we implemented and used in Algorithm 2. The first algorithm is not analyzed but it is very easy to show that in the limit $\gamma \to \infty$ it will be reduced to the overdamped Langevin MCMC we just discussed. Second algorithm is more complex in how friction $\gamma$ and inverse mass $u$ behave. *For comparison, we present results obtained by both algorithms, showing Algorithm 1 first.* As in the previous section, we start with $\delta = \sigma$; we set $\gamma = 1$ ($u = 1$ unless stated otherwise) with $\Delta k = 30$:

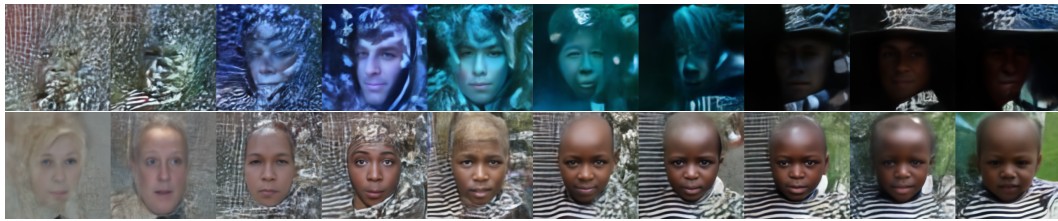

As in the previous section $\delta = \sigma$ is too high for FFHQ-256, $4 \otimes 8$ model. (However, note the relative stability of Algorithm 2, for this random seed.) Next, we consider $\delta = \sigma/2, \gamma = 1$, shown with $\Delta k = 100$:

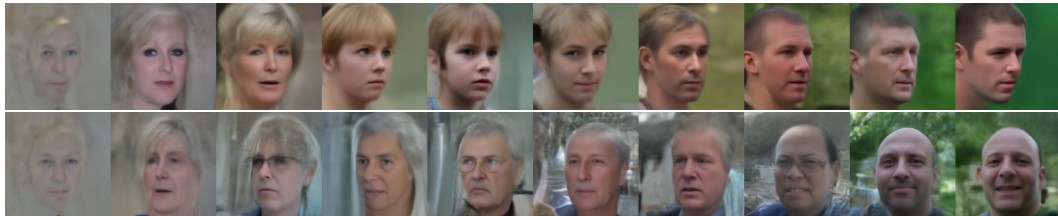

Note the remarkable contrast between the results above and the corresponding figure (for $\Delta k = 100$) in the previous section. We continue the chain with the same $\Delta k = 100$:

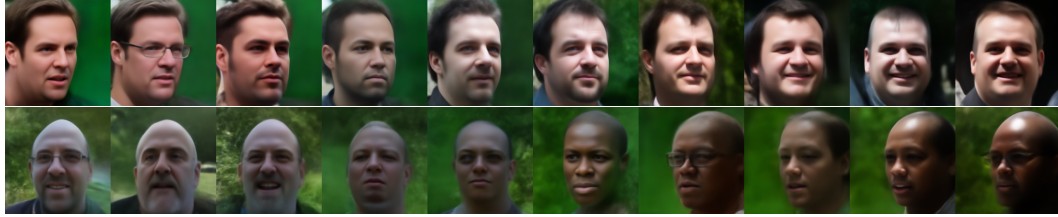

Both algorithms start to diverge for this choice of parameters ($\delta = \sigma/2, \gamma = 1$). To add more stability we can consider increasing the "mass" (lowering $u$). In (Cheng et al., 2018), $u$ is replaced with $1/L$, where $L$ is the Lipschitz constant of the score function. By lowering $u$, we are telling the algorithm that the score function is less smooth and MCMC uses more conservative updates. This can also be seen in the equations in both algorithms, where the step size $\delta$ is multiplied by $u$ in several places. However, note this effect is more subtle than just simply scaling $\delta$, *especially so for Algorithm 2* as can be seen by inspection. Next, we consider $u = 1/2$. This will affect the mixing time and the chains are now shown with $\Delta k = 1000$ (instead of $\Delta k = 100$ earlier):

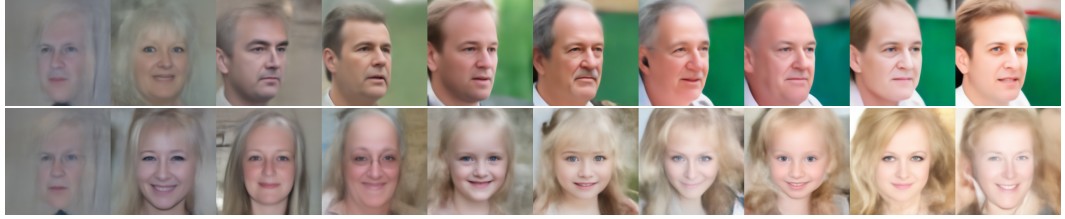

This experiment concludes this section. We come back to more FFHQ-256 chains in Appendix I.

# G   MNIST

## G.1   MORE IS DIFFERENT

There is a notion of congruence which arise in our analysis in Sec. 2.3 which we denote by "$\cong$": two different models, $\sigma \otimes M$ and $\sigma' \otimes M'$, "agree with each other" in terms of the plug-in estimator

$$\sigma \otimes M \cong \sigma' \otimes M' \text{ if } \frac{\sigma}{\sqrt{M}} = \frac{\sigma'}{\sqrt{M'}}.$$

However these models are vastly different, one is in $\mathbb{R}^{Md}$, the other in $\mathbb{R}^{M'd}$. And if $\sigma \gg \sigma'$ we expect that $\sigma \otimes M$ model to have better properties as a generative model. This is clear in the regime ($\sigma' \ll 1, M' = 1$). In that regime, almost no learning occurs: at test time MCMC either gets stuck in a mode for a long time (the best case scenario) or simply breaks down. We refer to (Saremi & Hyvärinen, 2019) for a geometric intuition on why large noise is essential in these models regarding the walk-jump sampling. We should point out that this is a general phenomenon in all denoising objectives (Alain & Bengio, 2014; Song & Ermon, 2020). In a nutshell, when noise is small the Langevin sampler finds it difficult to navigate in high dimensions and the forces of the Brownian motion eventually takes it to a part of space *unknown* to our model and MCMC breaks down (the first experiment below). If we are "lucky" we can find the manifold initially (the second experiment below) and MCMC may remain stable for a while but a model with $\sigma > \sigma'$ will have better mixing properties. We should highlight that in the analysis of Cheng et al. (2018) the mixing time scales as $\kappa^2$ where $\kappa$ is the *condition number* (see Sec. 1.4.1) proportional to the Lipschitz constant; this quantifies the intuition that MCMC on smoother densities (smaller Lipschitz constant) mixes faster.

To validate these intuitions we consider comparing $1 \otimes 16$ and $^1/4 \otimes 1$. In addition to statistically identical plug-in estimators, these models also arrive at similar training losses with the same training schedules (1.03 for $1 \otimes 16$ and 1.19 for $^1/4 \otimes 1$). We start with $^1/4 \otimes 1$. Below we show WJS using Algorithm 2 with the setting ($\delta = ^1/8, \gamma = ^1/2, u = 1$) (the setting $\delta = \sigma/2, \gamma = ^1/2, u = 1$ is a simple choice we used in testing all our $\sigma \otimes M$ models). Here, MCMC *breaks down* shortly after the last step shown here (this chain is shown with $\Delta k = 30$, in the remainder $\Delta k = 500$):

Algorithm 1 is more stable here and we arrive at the following chain (total of $10^5$ steps):

Now, for $1 \otimes 16$ model, using Algorithm 2 with parameters ($\delta = ^1/2, \gamma = ^1/2, u = 1$), we arrive at:

And using Algorithm 1 with the same parameters (and initial seed) we arrive at:

(Note the stark differences in terms of mixing between Algorithms 1 and 2 for this model.)

In summary, *more is different*: *quantitative differences become qualitative ones* (Anderson, 1972) and this is highlighted here for "congruent" $\sigma \otimes M$ models that differ in terms of the noise level. If computation is not an issue, one should always consider models with larger $\sigma$ but this remains a conjecture here in its limiting case (as $\sigma$ and $M$ grow unbounded, while $\sigma/\sqrt{M}$ remains "small").

## G.2 INCREASING $M$: TIME COMPLEXITY, SAMPLE QUALITY, AND MIXING TIME TRADE-OFFS

Here we extend the ablation study in the previous section by studying $\sigma \otimes M$ models with a fixed $\sigma$ where we increase $M$. In the previous section we studied two models where $\widehat{x}(\mathbf{y})$ has the same statistical properties as measured by the loss (and the plug-in estimator), and we validated the hypothesis that MCMC sampling has better mixing properties for the smoother M-density (with larger $\sigma$). Now, the question is what if we keep $\sigma$ fixed and increase $M$?

As we motivated in Sec. 1, by increasing $M$ the posterior $p(x|\mathbf{y})$ concentrates on the mean $\widehat{x}(\mathbf{y})$, thus increasing the sample quality in WJS. However there are trade-offs: (i) The first is the issue of *time complexity* which is an open problem here. In our architecture, the $M$ measurements are passed through a convolutional layer and after that models with different $M$ have approximately the same time complexity. The problem is this may not be an ideal architecture. Presumably, one should increase the network capacity when increasing $M$ (for the same $\sigma$) but by how much we do not know (this clearly depends on how complex $p_X$ is). Here we ignore this issue and we assume that our architecture has enough capacity for the largest $M = 16$ considered. (ii) The second trade-off is as we increase $M$, the price we pay for higher sample quality is that we will have longer mixing times, both in terms of generating the first sample, and more importantly in the mixing between modes. This trade-off is intuitive since $p(y_1, \ldots, y_M)$ becomes more complex for larger $M$ (for fixed $\sigma$). Below we compare WJS chains (as before, without warmup) for $\sigma = 1$ and $M = 1, 2, 4, 16$:

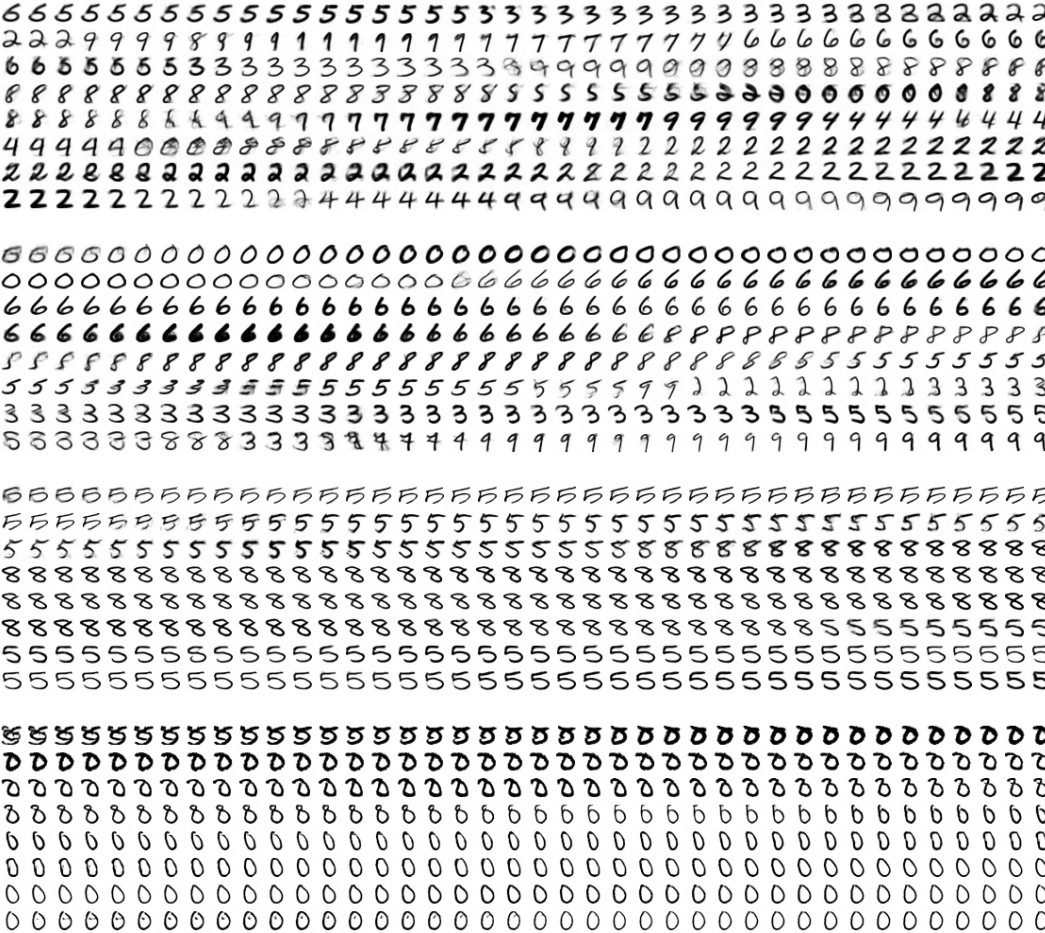

Figure 7: (*mixing time vs. image quality trade-off*) WJS chains for $1 \otimes M$ models are presented in "real time" ($\Delta k = 1$) starting from noise (320 steps in total), for $M = 1, 2, 4, 16$ in order from top pannel to the bottom one. We used Algorithm 2 ($\delta = 1/2, \gamma = 1/2, u = 1$) with the same initial seed. Note that, at the cost of sample quality, all classes are visited in $1 \otimes 1$ in the short chain presented.

### G.3   MUVB vs. MDAE

Below we make a one to one comparison between MUVB and MDAE for the $1 \otimes 4$ setting we trained on MNIST. In both cases we ran a single chain for **1 million+** steps. For sampling M-density we used the Langevin MCMC algorithm by Sachs et al. (2017) with parameters ($\delta = 1, \gamma = 1/4, u = 1$) (see Appendix D). *This is an "aggressive" choice of parameters for Langevin MCMC, designed for fast mixing, yet the chains do not break up to 1M+ steps (we stopped them due to computational reasons).*

Below we show the two chains at discrete time resolution of 5 steps (*the first sample, top left corner, is obtained after only 5 steps*). This comparison visually demonstrates that MUVB has better mixing and MDAE better sample quality. All digit classes are visited in MUVB in a variety of styles in 4000 steps shown here. MDAE chain is cleaner but the classes $\{0, 1, 5, 6\}$ are visited scarcely.

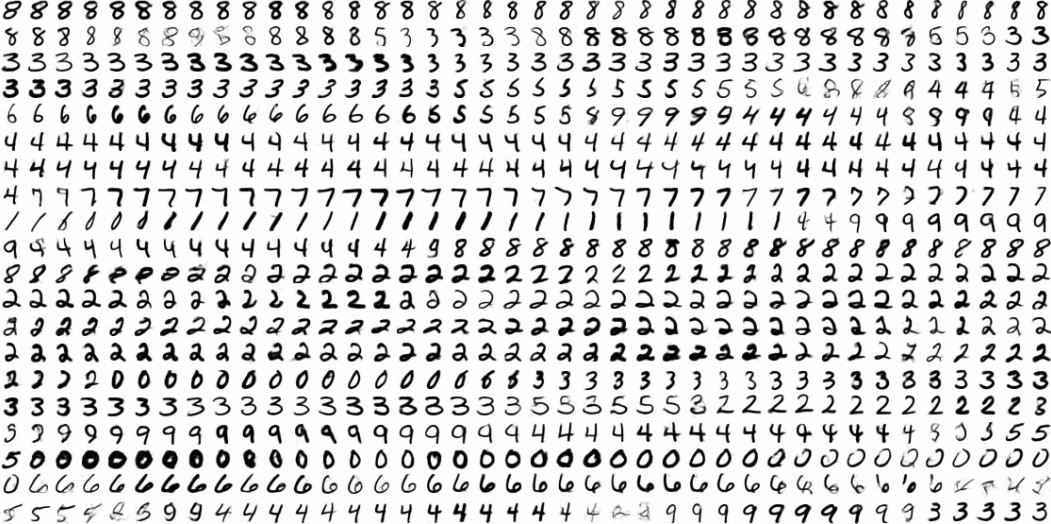

Figure 8: MUVB, $1 \otimes 4$ model on MNIST, $\Delta k = 5$ steps.

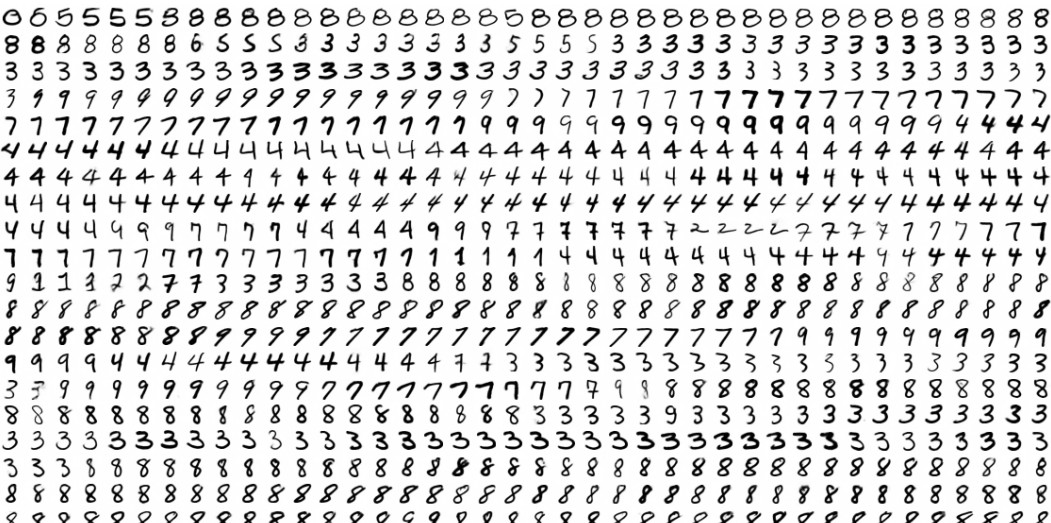

Figure 9: MDAE, $1 \otimes 4$ model on MNIST, $\Delta k = 5$ steps.

### G.4 LIFELONG MARKOV CHAIN

We demonstrate the WJS chain with 1 million+ steps obtained in MDAE $(1 \otimes 4)$ in its entirety (viewed left to right, top to bottom). We informally refer to these long chains that never break as *lifelong Markov chains*. The chain is reported here in its "entirety" to demonstrate the fact that it does not get stuck in a mode and remains stable through the end (we stopped it due to computational reasons).

Figure 10: MDAE, $1 \otimes 4$ model on MNIST, $\Delta k = 500$ steps, 1 million+ steps in total.

## H   CIFAR-10: SINGLE-CHAIN FID EVALUATION & SOME FAILURE MODES

The main "fear" in parametrizing the score function directly in MSM, which MDAE is an instance of, is we are not guaranteed to learn a conservative score function (see Sec. 4.1). More importantly, we do not have a control over how the learned model is failing in that regard. These fears were realized in our CIFAR experiments on the $1\otimes 8$ model presented below. The main results use Algorithm 2. Algorithm 1 simply fails for the Langevin MCMC parameters we experimented with below (for completeness we present a short chain at the bottom panel). In 2D toy experiments the non-conservative score functions were quantified in (Saremi et al., 2018, Fig. 1h), but this quantification is difficult in high dimensions. However, what is intriguing about the experiments here is that the "non-conservative nature" of the score function is manifested—this is merely a conjecture—in the *cyclic* mode visits apparent in the Markov chain.

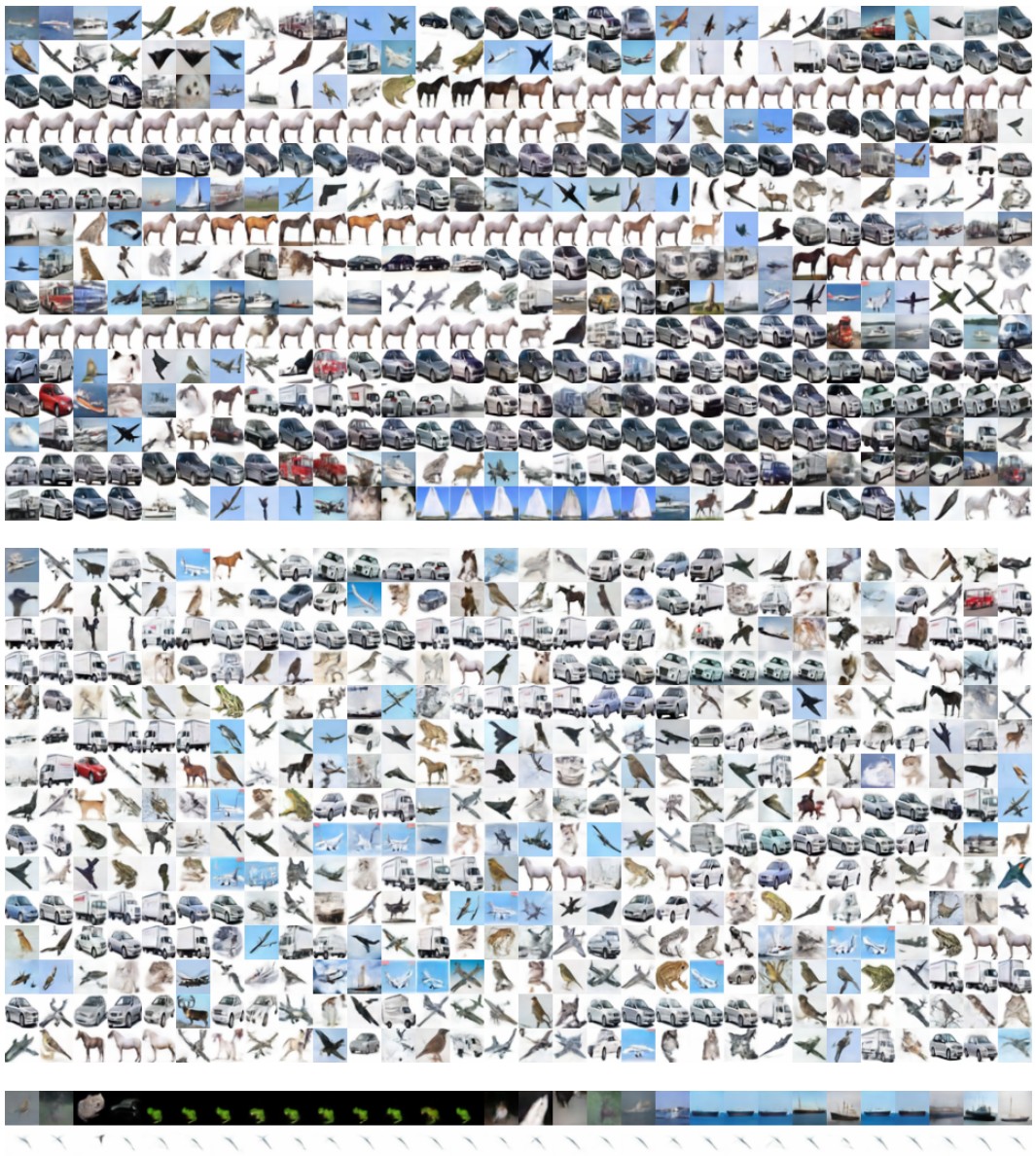

Figure 11: Top panels use Algorithm 2 at two different checkpoints. The bottom panel uses Algorithm 1. MCMC parameters are $(\delta = 0.1, \gamma = 2, u = 10)$ for the top and bottom panel, and $u = 20$ for the middle panel. In all cases $\Delta k = 500$ ($\approx 2 \times 10^5$ steps). The FID scores are respectively 91 & 79.

We encountered the same problem using *spectral normalization* (Miyato et al., 2018): the chain below was obtained using Algorithm 2 with the setting $(\delta = 1/2, \gamma = 1/2, u = 1)$ where we got the FID score of 99. We ran the chain for 1 M steps; the first 600 K steps are visualized below:

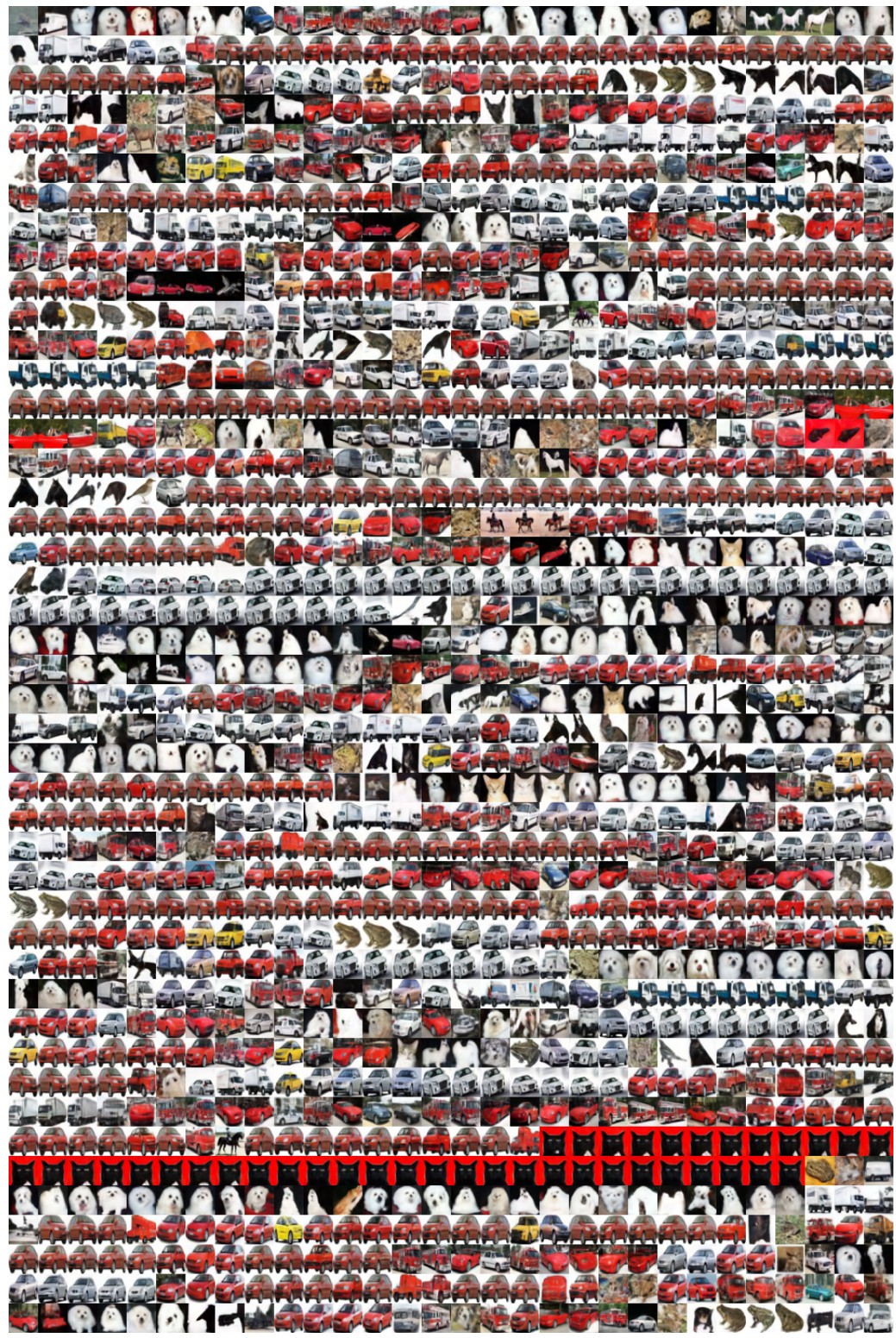

Figure 12: WJS chain for MDAE $1 \otimes 8$, trained with spectral normalization, $\Delta k = 500$.

We also experimented with training MDAE using SGD optimizer. Using Algorithm 1 with the setting $(\delta = 0.7, \gamma = 1, u = 1)$ the MCMC chain was stable for 1 M steps where we stopped the chain. We picked 50 K images at equal interval of $\Delta k = 20$ from the chain resulting in an FID of 43.95:

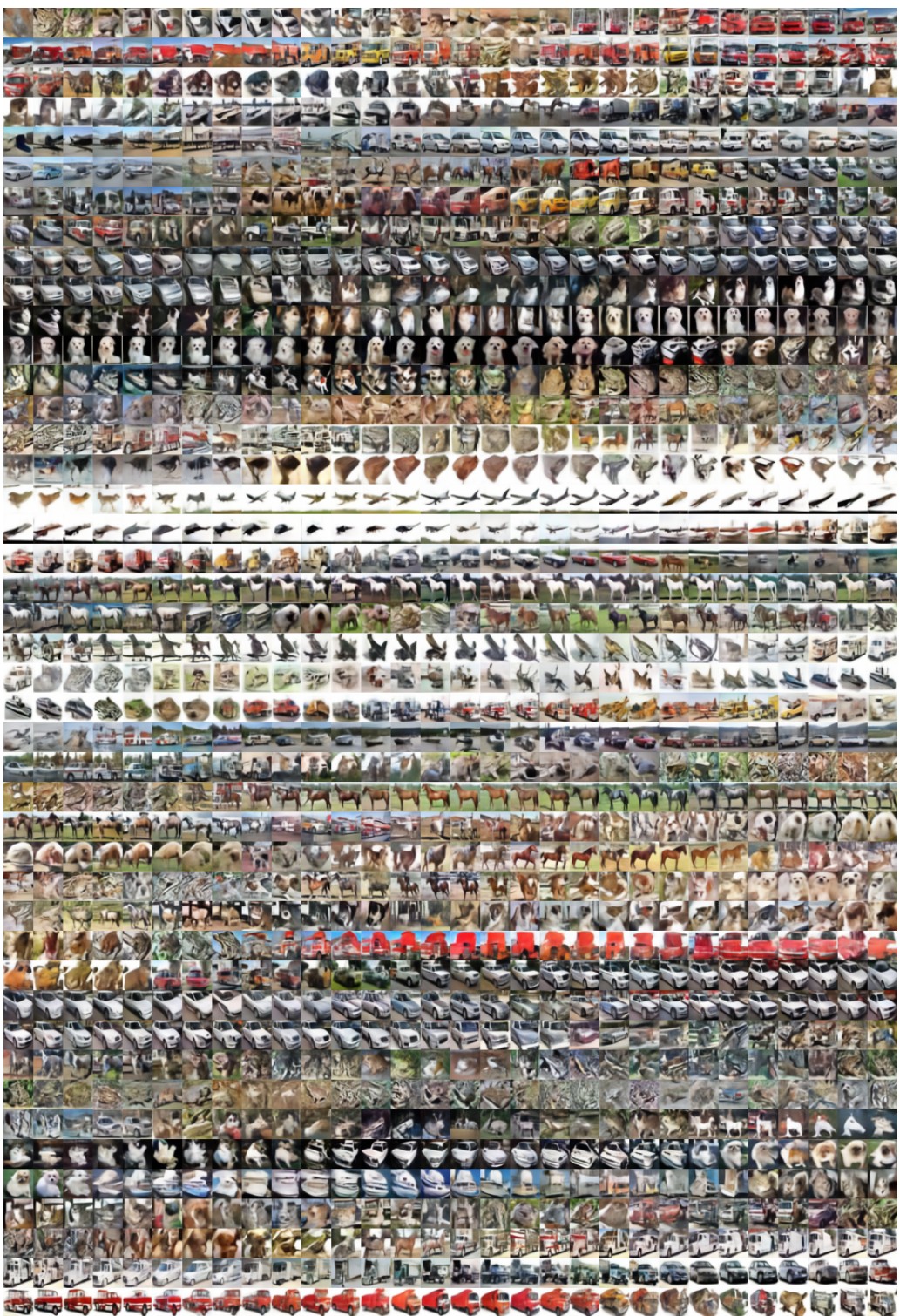

Figure 13: The first 13 K steps of a WJS chain with 1 M steps for MDAE $1 \otimes 8$ on CIFAR-10 with $\Delta k = 10$. We obtained the FID score of 43.95 by selecting 50 K samples skipping $\Delta k = 20$ steps.

We finish this section with a discussion on numerical comparisons on the sample quality between our MDAE $1 \otimes 8$ model and other recent MCMC-based methods, summarized in Table 4:

- The table excludes denoising diffusion models since the MCMC chain in such models are "conditional" in the sense that sampling is via reversing a diffusion process based on a *sequence of conditional distributions* which is learned during training (see the discussion in Sec. 7). By comparison, in all the papers in the table below there is only one energy/score function that is being learned.

- The use of the term "long chain" below is based on the current status of the field, chains of order 1,000 steps: the MCMC chains used for generation and reporting FID scores in most prior works has been "short", of order 50 to 100 steps by comparison which we have indicated in a column in the table. The FID scores in those papers were obtained by 50 K *short-run parallel chains*.

- Our work is unique in this literature in that for the first time we report competitive FID scores obtained from a *single MCMC chain*. As we emphasized in Sec. 6 the quest for generating very long "life long" chains with good sample quality and mixing properties, as measured by the FID score (Heusel et al., 2017), is a scientific challenge and we believe it is an ultimate test to demonstrate that one has found a good approximation to the true energy/score function. We should point out that the only competitive paper here in obtaining FID scores with long MCMC chains is by Nijkamp et al. (2022) where the "long chains" have been limited to 2,000 steps and the FID obtained is much worse (78.12) than what we obtained (43.95) with our $1 \otimes 8$ model from a single MCMC chain of 1,000,000 steps (see Fig. 13).

Table 4: FID results for unconditional MCMC-based sample generation on CIFAR-10

|  | FID | long chain | single chain | MCMC steps |
| --- | --- | --- | --- | --- |
| Xie et al. (2018) | 35.25 | ✗ | ✗ | N/A |
| Nijkamp et al. (2019) | 23.02 | ✗ | ✗ | 100 |
| Du & Mordatch (2019) | 40.58 | ✗ | ✗ | 60 |
| Zhao et al. (2020) | **16.71** | ✗ | ✗ | 60 |
| Xie et al. (2021) | 36.20 | ✗ | ✗ | 50 |
| Nijkamp et al. (2022) | 78.12 | ✓ | ✗ | 2,000 |
| MDAE, $1 \otimes 8$ (our work) | 43.95 | ✓ | ✓ | **1,000,000** |

# I  FFHQ-256

In this section, we provide several WJS chains for our MDAE $4 \otimes 8$ model on FFHQ-256 dataset. We refer to Algorithm 1 by **A1** and Algorithm 2 by **A2**, and MCMC parameters are listed as $(\delta, \gamma, u)$.

A1; $(2, 1, 1)$; $\Delta k = 300$

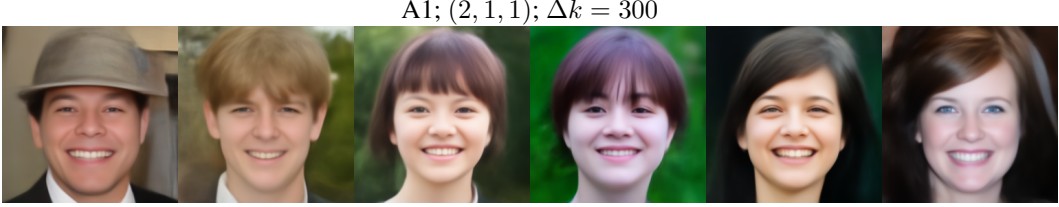

A2; $(2, 1, 1)$; $\Delta k = 200$

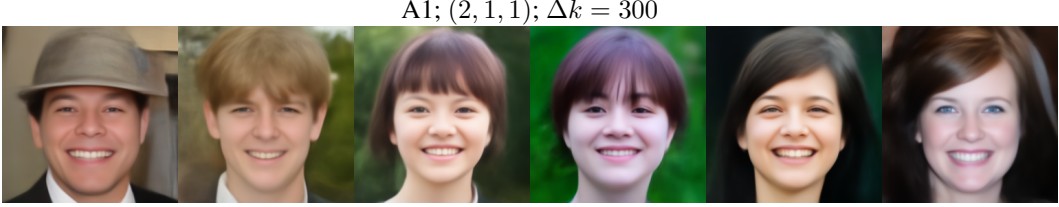

A2; $(^1/_{10}, 1, 10)$; $\Delta k = 500$

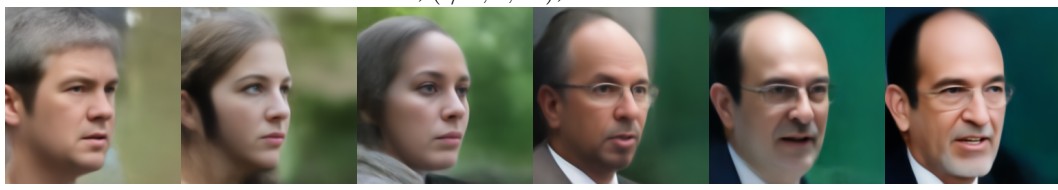

A2; $(4, ^1/_2, ^1/_4)$; $\Delta k = 150$

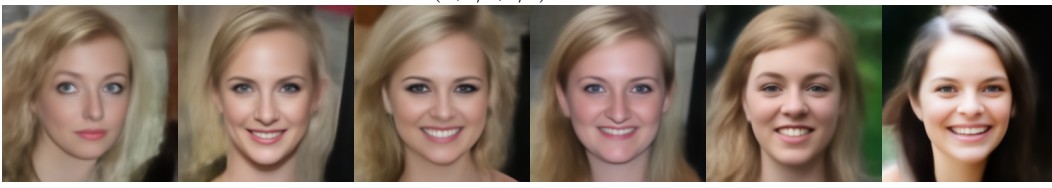

A1; $(2, ^1/_2, 1)$; $\Delta k = 100$

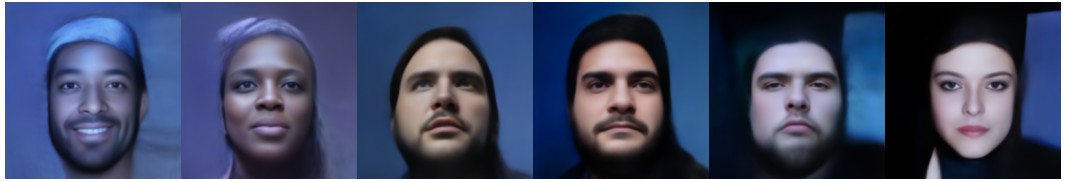

A1; $(2, ^1/_2, 1)$; $\Delta k = 20$

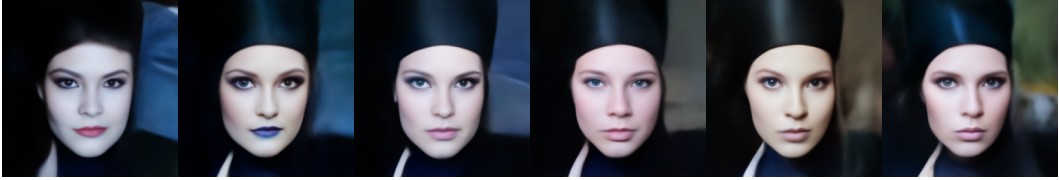

