# OpenReview forum: "Multimeasurement Generative Models"
_ICLR.cc/2022/Conference — ICLR 2022 Poster_

### Official Review · Reviewer_6P94 · 2021-10-31

**Correctness:** 1
**Technical Novelty And Significance:** 1
**Empirical Novelty And Significance:** 3
**Recommendation:** 8
**Confidence:** 4

**Main Review:**

${\bf \text{Strengths}}$

1-The problem addressed in this paper (generating new independent samples of a given selection of samples drawn from an unknown distribution) is of great interest to the machine learning community.

2-The approach used (going through a variety where the density estimation is easier) is very original. The intuitions and connection with literature are interesting and allow a new point of view of the algorithms already used (DAE as an example). Several experiments have been designed to justify the relevance of the approach.

3-The article is quite well written and well structured.

4-Although the algorithm uses already known sampling methods and energy function optimization methods, the general idea of the Bayesian estimator is an interesting contribution to the machine learning community.

${\bf \text{Weaknesses}}$

1-The motivation of presenting the Poisson MNM is unclear specifically since it is never used afterward and more importantly no motivation to discard it and prefer the Gaussian MNM. More specifically, isn't it to present a simpler version of MNM before tackling the more involved Gaussian MNM? But more importantly what are the disadvantages of Poisson MNM to discard it?

2-The discussion of the higher noise level is also not convincing. Moreover, the advantage/disadvantage of increasing M is not clearly discussed (only the advantages are pointed out). I am quite convinced that there is a great connection between $M$ and $\sigma$. More concretely, there is no discussion about the time complexity of the algorithm which may be related to increasing $M$. It seems clear that with a low level of noise we do not need to have too many channels measurement (independent samples). Therefore for me, low values of $\sigma$ and low value of $M$ are comparable to high values of $\sigma$ and high value of $M$. So in the comparison for MNIST (between two different values of $\sigma$), it is not fair to change the value of $M$. In an ablation study, you normally need to keep $M$ fixed and change $\sigma$ since the two are connected.  Having big values of $\sigma$ is great to smooth well the manifold but in compensation, we need high values of $M$, which may be problematic for time complexity (which is not discussed).

3-In the abstract, I have the impression that the sentence "Samples from $p(x)$ are obtained by walk jump
sampling, using underdamped Langevin MCMC to sample from $p(y)$" is incomplete since we should close the loop and mention that the empirical Bayes estimator is used after this underdamped Langevin.

4-I the abstract, it is mentioned fast mixing. But the term fast is never concretely shown (no running time,...)

5-In the concentration rate of the empirical estimate, it is mentioned that the optimal estimator is expected to give a better rate than the rate of Tao. This is not proved, right? More concretely what is the rate of the optimal estimator? What is the intuition to say that the rate will be better than just a constant?

6-In the definition of the loss $\sum_{m=1}^{M}\mathcal{L}^{m}(\theta)$, why a normalization with $\frac 1M\sum_{m=1}^{M}\mathcal{L}^{m}(\theta)$ is not used since the large $M$ case is included in the framework? More concretely is the limit $\lim_{M\rightarrow\infty}\sum_{m=1}^{M}\mathcal{L}^{m}(\theta)$ correctly defined?

7-In section Denoising score matching it is written 'empircal' instead of empirical


**Summary Of The Paper:**

Given $n$ independent samples $x_i$ in a space of dimension $d$ drawn from an unknown distribution $p(x)$, this paper is interested in drawing new samples independent of $x_i$ but coming from the same unknown distribution p(x).

The classical approach consists of learning $p(x)$ from $x_i$ (i.e., approximated by $\tilde{p}(x)$) and sampling new points according to the approximated version $\tilde{p}(x)$, which remains a difficult task given sometimes the highly non-convex character of $p(x)$.

For these reasons, the authors propose to noise the data $x$ using $M$ different channels whose noise levels are chosen to obtain data denoted $y$. Since the level noise of each channel is known, this allows using a Bayesian estimator $\hat x(y)$ to find the samples in the original space. The advantage of this approach is that the Bayesian estimator depends on the density $p(y)$, which allows, thanks to a chosen parameterization of $p(y)$ (which therefore leads to a parametrization of $\hat{x}(y)$), to find the optimal parameters through a least-square objective by minimizing $\|x-\hat{x}(y)\|^2$ on the training data. The Bayesian estimator thus allows generating new samples using the optimized density $p^{\star}(y)$ with the corresponding optimal parameters.

The authors make a connection between the proposed method and some methods of the literature constituting new intuitions, different views of algorithms proposed in the literature. The experiments conducted suggest the efficiency of the algorithm to generate the most diverse samples

**Summary Of The Review:**

The paper proposes an original work by offering a way out of the problem of approximating very complex densities by passing through a smoother variety. The method can be inspiring since it makes some connection and gives another point of view of the Denoising AutoEncoder proposed in the literature. The experiment shows that the method allows for generating stable and mixed samples and does not get stuck to some mode. Even if some discussions are not well-motivated, I recommend acceptance for the paper.

---

> ### Author Response · Authors · 2021-11-12
> **Response to Reviewer 6P94**
>
> Thank you for your comments. Your comments in the Main Review are addressed in order below:
>
> 1) Regarding the Poisson MNM, please see the response addressed to all reviewers.
>
> 2) The reviewer has correctly observed that $\sigma$ and $M$ are connected. This motivated us to compare models with the same _effective_ $\sigma$ in G.1 (which result in similar training loss), and check if they behave differently or not. We fully agree that clear comparisons where $\sigma$ or $M$ are kept fixed are also important. We already have these results, e.g. MNIST 1$\otimes$4 (Fig. 8) vs 1$\otimes$16 (Sec. G.1) and we will present them as a proper comparison in the revision. The issue of time complexity of increasing M is intriguing but quite complex. In our architecture the M measurements are passed through a convolutional layer and after that models with different M have approximately the same time complexity. However, the issue is this is probably not an ideal architecture. Presumably, one should also increase the network capacity when increasing M (for the same $\sigma$) but by how much we do not know. We will include a discussion of this issue in the appendix.
>
> 3) The sentence was shortened due to space—the multimeasurement Bayes estimation (jump) was implied—but we will edit that sentence in light of your comment. Thank you for the comment.
>
> 4) We will clarify what we mean by using “fast mixing” to describe our results. We use it to refer to two properties of interest: in how many MCMC steps does the chain start producing samples from the target distribution after initialization (called “mixing time”), and how quickly does the continued chain mix between modes of the distribution (we can call it “mode mixing time”). The mixing time for our MNIST 1$\otimes$4 model is about 5 steps (Figs. 7 and 8), while for FFHQ-256 is about 100 steps (Fig. 3). For MNIST we are not aware of any model with such a small mixing time, and there are no unconditional MCMC results for 256x256 in the literature, though Nijkamp et al. (2019) did report a mixing time of about 100 steps for 128x128 images to obtain the first sample, after which their chains tend to get “stuck”. Quantifying mode mixing time is much more difficult, since no defined metrics exist and long MCMC chains in high dimensions have not been reported previously. But we can make a qualitative observation based on the obtained chains that MCMC does not get stuck and traverses various modes of the data distribution. Regarding running time that you specifically asked, as an example the FFHQ 4$\otimes$8 sequence in Fig. 3b (1000 steps in total) were obtained in shortly under 2 minutes (1:57) on V100.
>
> 5) We do not have a theoretical result on the concentration of the optimal estimator. Regarding your question “What is the intuition to say that the rate will be better than just a constant?” If you’re referring to the sqrt(d/M) rate, it is indeed possible that the optimal (Bayes) estimator follows the same rate but it is the “constant” that is vastly different between the two since the plug-in estimator completely ignores the correlations between noisy pixels (within and across noise channels).
>
> 6) We dropped the 1/M factor since it does not affect the optimal solution and for having cleaner equations since we have it again showing up in Eqs. 17 and 19 (in that page we do not have any space left!). However, in light of your question we plan to add 1/M to Eq. 14 and will try to work around the space issue in Sec. 4.
>
> 7) Thank you, it will be fixed.
>
> We hope we answered your concerns but we’d be happy to elaborate more if there are follow up questions. We plan to incorporate your comments and our response here in an update to the manuscript soon.

---

### Official Review · Reviewer_mPcN · 2021-11-02

**Correctness:** 3
**Technical Novelty And Significance:** 3
**Empirical Novelty And Significance:** 3
**Recommendation:** 6
**Confidence:** 3

**Main Review:**

This paper combined DAE and empirical Bayes to learn and sample from some unknown distribution. This method is a novel method and is very different from previous methods that use convolution to smooth the distributions. While other methods are based on weighted noise or annealing, different noise levels are equal in this method.

The main body of the paper is in general well-written. I have the following questions on the paper.
1. Sec 2.1 derived the formula for poisson MNM, but it seems the other parts of the paper are mainly based on Gaussian NMN. Are there any further results on Poisson MNM? How does it compare with the Gaussian MNM?
2. Sec 2.3 gives a concentration rate of the estimator, but it is not straightforward to see how sigma_eff is derived.
3. Sec 5 mentioned that underdamped langevin is better than langevin because UL scales as sqrt d theoretically. It is also mentioned in the appendix that in your experiments, UL mixes faster. Are there any experimental results on how much faster it mixes?


**Summary Of The Paper:**

This paper proposed a multimeasurement noise model to learn and sample from some unknown distribution and demonstrated its effectiveness on datasets like MNIST, CIAR-10 and FFHQ-256. It studies the convolution of the distribution with different levels of noise and uses the Bayes estimator in each noise level to recover the original sample.

**Summary Of The Review:**

This paper is in general a good paper. The new generative method it proposed can be of interest to the ICLR community.

---

> ### Author Response · Authors · 2021-11-12
> **Response to Reviewer mPcN**
>
> Thank you for your comments. Your questions are addressed below which we plan to incorporate in our update to the paper:
>
> 1) Regarding the Poisson MNM, please see the response addressed to all reviewers.
>
> 2) The calculation for $\sigma_{\rm eff}$ is based on the fact that $y_m = x + \varepsilon_m$, where $\varepsilon_m \sim N(0,\sigma_m^2 I_d)$. It follows $x-M^{-1} \sum_m y_m=-M^{-1} \sum_m \varepsilon_m$ which has the same law as $N(0,\sigma_{\rm eff}^2 I_d)$. We can then readily use the concentration results for Gaussian distributions. Thank you for the question. We will add this explanation in our update to the manuscript.
>
> 3) The experimental results are given in comparing overdamped and (the two versions) of underdamped Langevin MCMC are given in Appendix F. The differences are significant: in overdamped Langevin MCMC we do not get a sample after 1000 steps (the figure after “Now consider $\delta = \sigma/2$ and $\Delta k = 100$” in Appendix F) while in 1000 steps we can generate a whole sequence with many modes explored with underdamped Langevin MCMC (see Fig. 3b).

---

> > ### Comment · Reviewer_mPcN · 2021-11-30
> > **Response**
> >
> > Thank you for your response. I have read all the reviews and comments. I decide to keep my score.

---

### Official Review · Reviewer_dnHy · 2021-11-02

**Correctness:** 4
**Technical Novelty And Significance:** 3
**Empirical Novelty And Significance:** 2
**Recommendation:** 6
**Confidence:** 4

**Details Of Ethics Concerns:**

Generative models by the nature of its training procedure produce samples that are similar or identical to some samples in the training set. Therefore, extra care should be taken to abide the ethical principles when using generative models. The authors pointed out Prabhu & Birhane 2020 for in-depth discussions on this subject, and the usage of the data in this paper follows required licenses.

**Main Review:**

1. The author claim that the smoothed M-density $p_\mathbf{Y}$ is easier to learn from than the original distribution $p_\mathbf{X}$ in high-dimensional settings. However, there are no simple examples or existing literature included in the manuscript to support this statement. The difficulty of sampling from a high-dimensional distribution is a core problem in designing generative models, so the authors should consider making this motivation clear and better supported.
2. In the introduction, the authors stated that the MCMC is considered an art that cannot converges fast for complex distribution, but later in Section 5, the authors adopted Langevin MCMC to sample the M-density. It is encouraged to compare the MCMC with the Langevin MCMC for M-density since the sampling efficiency is one of the selling point of this paper. For example, is the convergence rate of Langevin MCMC faster? The authors should at least show it in the empirical study.
3. The authors did not provide a good motivation for why considering the Poisson and Gaussian MNMs in Section 2. What about other MNMs? Does the choice of the MNM kernel related to the properties of $p_\mathbf{X}$? Is there an optimal MNM kernel to pick? In Section 2 the authors focused on the derivation of the closed-form expressions of the MNMs, but the reasoning on why they chose them is more important in my opinion.
4. The authors proposed several parameterization schemes in Section 4, including MDAE, MEM${}^2$, and MUVB. What are the pros and cons of them over each other? A discussion is needed or at least added in Appendix.
5. The permutation invariance is one the appealing points of using MNMs, and in the Introduction, the authors stated ``In Sec. 6, we present our experiments on MNIST, CIFAR-10, and FFHQ-256 datasets which were focused on permutation invariant M-densities.''. However, in Section~6, the discussion on the permutation invariant M-densities is missing.
6. An empirical comparison between MDAE and other generative models is lacking. How to compare the quality of the generated samples? In the illustrations, the authors focused on facial images, and in some figures, the hair or background in the images are not generated well, why? It is unclear about the advantages of the proposed MDAE over other generative methods, and a discussion is needed.
7. The abbreviation DAE for denoising auto-encoder is used without definition in the abstract. Although DAE is well-known in the field, it is suggested that the authors still defined all the abbreviations before usage in this manuscript. Another example is MCMC for Monte-Carlo Markov chain.

**Summary Of The Paper:**

This paper introduced an alternative sampling method, with an application on generative models, by convolving an unknown distribution $p_x$ with a factorial kernel called multi-measurement noise model (MNM). The resulting M-density $p_y$ is smoother (easier to sample from), and is permutation invariant. Two factorial kernels, Poisson and Gaussian MNMs, are introduced for the convolution, and can be connected to Bayes estimator, and the learning of parametric energy and score functions. Two parameterization schemes are proposed for modeling the energy and score functions of the Gaussian M-density respectively. Empirical results on FFHQ-256 dataset are very impressive. The main contribution is the usage of factorial kernels, which is a direct extension of smoothing a density in non-parametric density estimation, and is not used for generative models before.

**Summary Of The Review:**

The proposed MNMs and MDAE are very appealing, and novel. I believe this will be a great publication if the authors could please address the issues listed above. In particular, I think the authors should give (1) a better motivation on why this methodology outperforms other generative methods, (2) how to compare different parameterization schemes, (3) a more thorough discussion on the experiments regarding the permutation invariance, efficiency over MCMC or other methods, and (4) companion on the quality of the generated samples with existing methods.

---

> ### Author Response · Authors · 2021-11-12
> **Response to Reviewer dnHy**
>
> Thank you for your detailed comments. Your comments in the Main Review are addressed in order below:
>
> 1) We are not sure if the reviewer is referring to the difficulty in learning or sampling (the first part of the comment referred to learning, the later part to sampling). Regarding learning, we already had a discussion in the third paragraph in the introduction. There is a thin literature in nonparametric density estimation literature on this topic but the paper we cited is a strong paper on estimating smoothed densities. We believe it is a great research direction to extend their results to M-densities (in particular to see how $M$ shows up in the convergence rate). Regarding sampling, theoretically we only know the analysis for log-concave densities (convex energy functions), e.g. in Cheng et al. (2018) their rates of convergence scales as $\kappa^2$ where $\kappa$ is proportional to the Lipschitz constant of the score function, therefore smoother densities (with smaller Lipschitz constant) lead to shorter mixing time. Of course, this is also expected intuitively beyond log-concave densities. Cheng et al. (2018) is already cited in the paper but we will include a specific reference regarding how their rate of convergence scales with the Lipschitz constant to better motivate our approach. This will indeed improve the paper and we are thankful to the reviewer for their question.
>
> 2) We think our phrasing around this may have led to a misunderstanding for the reviewer. By “MCMC” we refer to the general class of MCMC algorithms, of which “Langevin MCMC” is a subset. Our comment regarding MCMC being an art that is difficult to use for complex high-dimensional distributions applies to all MCMC methods, including Langevin MCMC (which is why there are no successful examples of it in the literature for complex distributions in high dimensions). We would like to highlight that in Appendix F we did have comparisons between different Langevin MCMC algorithms.
>
> 3) Regarding Poisson MNM please see the response addressed to all reviewers.
>
> 4) We will be happy to add further discussion on this to the appendix. As mentioned in Sec.4, the MEM family is more theoretically sound (its score function is a proper gradient field). However, it increases the  computational burden since we need two backward passes during learning: we need a backward pass through the score function (to compute the gradient of the loss) which itself is computed with a backward pass. For Langevin MCMC, models in the MEM family require a backward pass to compute the score function (again at the cost of computation time) at each iteration of the algorithm (compared to the MSM family that requires none). Appendix G.2 was devoted to comparing MUVB (in the MEM family) with MDAE (in the MSM family). Our comparison is visual in showing the MUVB mixes faster than MDAE but  MDAE gives rise to sharper samples. At present, we do not know why energy models like MUVB (that require two backward passes) are worse regarding sample quality than score models like MDAE. This is a deep question that has not been addressed in the literature and we do not yet have an answer to.
>
> 5) The main appeal of permutation invariance regarding Langevin MCMC is that we can set the MCMC parameters $(\delta, \gamma, u)$ to be the same for all noise channels due to the permutation symmetry. We will highlight this point regarding the permutation symmetry and Langevin MCMC in Sec. 5.
>
> 6) In experimental comparisons, our main goal in this paper was to demonstrate long stable fast mixing chains as a scientific challenge as we discussed around “lifelong Markov chains” in Sec. 6. Regarding sample quality comparisons with the literature on MCMC chains, we highlight that there are no studies on 256x256 resolution images (and our results are remarkable in that sense as you mentioned). At present we do not expect the lack of sharpness you observe to be a fundamental limit of our models, but further work is needed to scale up our models with improved architectures, large $M$, and perhaps better sampling algorithms. Please see also our response to all reviewers where we highlight a submission to this conference that also attempts to achieve long MCMC chains.
>
> 7) Thanks for catching this. We will fix it in an update soon.
>
> We hope this answered your concerns but we’d be happy to elaborate more if there are questions left. We plan to incorporate your comments and our response here in an update to the manuscript soon.

---

> > ### Comment · Reviewer_dnHy · 2021-11-29
> > **Feedback**
> >
> > I appreciate the detailed answers from the authors.
> >
> > 1. I think the explanation should be added in the Introduction to better help readers to understand.
> > 2. Thanks for pointing out the clarification in Appendix F.
> > 3. As the Poisson MNMs are only a warm-up example for MNMs, it is suggested to move the Poisson MNMs for readers' information and to avoid digression.
> > 4. I still think that the discussion on time complexity is not very clear, as the fact that smoother distributions are easier to sample is the main selling point of this paper.
> > 5&6. It is better when the authors clearly said that "our main goal in this paper was to demonstrate long stable fast mixing chains as a scientific challenge". In the original writing, I think the connection to generative model is over-emphasized.
> > 7. Thanks.
> >
> > I think the authors did a good job for the rebuttal in general. However, the message that should be conveyed by this paper is still not crystal clear, in particular on the complexity. I decided to maintain my score but I will be happy if the paper gets in.

---

### Author Response · Authors · 2021-11-12
**Response to all Reviewers**

We are very grateful for the detailed and valuable comments from all reviewers, and glad to see that all reviewers agree that our results are novel and interesting. We are working on incorporating all suggestions into an upcoming revision, but meanwhile we will provide our responses to the common questions here.

#### **On Poisson MNMs**

Regarding motivation for the Poisson derivation, we recognize that this was a common concern among all reviewers, so we address it here:

- We included Poisson MNM for its simplicity and as a “warm up” to introduce the new formalism especially since there are commonalities between Poisson and Gaussian MNMs. Our example in page 4 is valuable since it demonstrates that the Bayes estimator is invariant to the noise channel one chooses to compute it. In comparison, the corresponding example for Gaussian MNMs (in Appendix A) is much more involved.

- The difficulty with Poisson MNM is that during sampling we should sample a discrete distribution and we simply do not know how to do it efficiently in high dimensions, unlike the Gaussian case where we can readily use Langevin MCMC (note that the score function is not well defined for discrete distributions). We can use Gibbs sampling for the Poisson case but it is very slow to mix (and one reason Boltzmann machines have not been very effective for modeling complex high dimensional distributions).

- Regarding the question “Is there an optimal MNM kernel to pick?” the short answer is we don’t know. Ideally, in addition to the sampling concerns above, we like an MNM such that the Bayes estimator is more concentrated at clean data. Regarding sampling, we currently only know how to do it efficiently with Gaussian noise models, since the Bayes estimator only depends on the score function which we learn and we can readily use Langevin MCMC. There is plenty of recent research on Langevin MCMC algorithms where researchers have pushed the dimension dependence of the mixing time from which our model benefits greatly (as we expand the dimension M-fold), so in that sense Gaussian MNM is the optimal MNM kernel at the present time. This is directly related to the discussion we have in Sec. 5.

We will write a summary of this explanation in the paper or in the appendix depending on space.

#### **A related parallel work**

Relevant to the question regarding empirical comparison by Reviewer dnHy, we would like to point out that in preparing this response, we became aware of a paper under submission to ICLR 2022 titled [MCMC Should Mix: Learning Energy-Based Model with Flow-Based Backbone](https://openreview.net/forum?id=4C93Qvn-tz) where they model $p_X$, along the lines of (Nijkamp et al., 2019; Du & Mordatch, 2019; Du et al., 2020) that we have cited in Sec. 6. In this recent submission they attempt to push “very long chains” to 2000 step chains. In that paper, their large scale model has been limited to CelebA 64x64 dataset. We encourage you to compare our results with the ones reported there (Figs. 3, 5, 10, and 11 for example). This is not meant as a criticism of their work as the methodologies are vastly different, but the comparison shows how superior our current model is in terms of sample quality, mixing time, and the stability of MCMC chains.

---

### Author Response · Authors · 2021-11-18
**Revised Manuscript**

We have uploaded the revised manuscript in which we have integrated our response to all reviewers. We believe that the new version addresses all the important points that were raised, but we will be happy to make additional changes to address any remaining concerns.

- In the main paper, the revisions have been minor but we believe they are important and we’re thankful to all reviewers for their questions.

- All reviewers had questions about the **Poisson MNM**: we expanded the discussion in the introductory paragraph in Sec. 2 to clarify why we included the theoretical results and why we didn’t study WJS on Poisson M-densities in our experiments, and we added Remark 7 in Appendix D to further expand on it

We summarize further revisions below which we organize by comments from specific reviewers:

#### **Reviewer dnHy**

- The abbreviation DAE is removed from the abstract, and we also use the full expression Markov chain Monte Carlo in the first paragraph; there were very minor edits to make it fit.

- In caption to Fig. 2, we emphasized the benefit of permutation invariant M-densities regarding MCMC sampling. For learning, we do not have analytical results on permutation invariance (which is simply a choice in our formalism) and we refrained further speculations on its benefits. We do believe this is an important direction to explore in future research regarding both learning and sampling.

- We have added further discussion in Appendix G.1 on the benefits of large noise for MCMC sampling, citing Cheng et al. (2018) on how mixing time scales with the Lipschitz constant of the score function.

- More related to some questions by Reviewer 6P94, but we added Sec. G.2 with an ablation study that is also a response to some of your questions regarding the issue of image quality.


#### **Reviewer mPcN**

We expanded Sec. 2.3 (below Eq. 10) to answer your question on the derivation of $\sigma_{\rm eff}$.

#### **Reviewer 6P94**

- We expanded the sentence in the abstract regarding walk-jump sampling.

- In Appendix G.1, we clarified our motivation for comparing models with the same $\sigma_{\rm eff}$ by stating that, in addition to these models having identical plug-in estimators, the training losses obtained for both models are also similar, which we now report.
- We removed the fifth paragraph and extended the sixth paragraph in Sec. 6, where we added a reference to the new appendix section (G.2) on the issue of time complexity that you raised.
- In Appendix G.2, we also compare $\sigma \otimes M$ models for fixed $\sigma=1$ as you specifically asked, and discuss the sample quality vs. mixing time trade-offs. This is indeed a good addition to the paper, and we are thankful to you for suggesting it.

- We brought (back) the 1/M factor to Eqs. 14, 17, and 19. There were some very minor edits in Sec. 4 to work around the space issue.

---

### Decision · Program_Chairs · 2022-01-20

**Decision:**

Accept (Poster)

**Comment:**

This paper studies the problem of generative modeling by convolving an unknown complex density with a factorial kernel called multi-measurement noise model (MNM) to obtain a smoother density that is easier to sample from. Poisson and Gaussian MNMs are proposed for the convolution. Experiment regarding image synthesis are conducted to demonstrate the effectiveness of the proposed framework. The paper studies a problem that is of great interest to the machine learning community, and the results are impressive and promising. However, the paper in the current form lack a comparative discussion and a quantitative comparison with some related works. After the rebuttal, all three reviewers tend to accept the paper. After several rounds of internal discussion among AC, reviewers and authors, the AC agrees with the reviewers, and recommends accepting the paper, given the changes the authors promised to make.

In summary, the AC recommends an acceptance and urges the authors to further revise their paper by adding a comparative discussion with those closely related works regarding generative models using MCMC sampling.